



# Validation of TROPOMI and WRF-Chem $NO_2$ across seasons using SWING+ and surface observations over Bucharest

Antoine Pasternak[1], Jean-François Müller[1], Catalina Poraicu[1], Alexis Merlaud[1], Frederik Tack[1], and Trissevgeni Stavrakou[1]

[1]Atmospheric Composition Department, Royal Belgian Institute for Space Aeronomy (BIRA-IASB), Avenue Circulaire 3, 1180 Brussels, Belgium

**Correspondence:** Antoine Pasternak (antoine.pasternak@aeronomie.be)

**Abstract.** Nitrogen oxides ($NO_x$) are key pollutants involved in ozone and particulate matter formation, with strong spatial variability near urban sources. Accurate monitoring of tropospheric nitrogen dioxide ($NO_2$) is essential for air quality management and relies on validated chemistry transport models and multi-scale observations. This study evaluates the WRF-Chem model v4.5.1, run at 1 km resolution over Bucharest, Romania, using in situ meteorological data and surface chemical measurements, as well as airborne $NO_2$ columns from 17 SWING+ flights conducted between 2021 and 2022. The model successfully captures key atmospheric processes and $NO_2$ variability across all but one observation period. Our results indicate that anthropogenic $NO_x$ emissions from CAMS-REG v7.0 are underestimated, with satisfactory agreement with observations achieved when the emissions are scaled by a factor of 1.5. We also assess TROPOMI tropospheric $NO_2$ columns v2.4.0 using SWING+ as reference, with WRF-Chem used as an intercomparison platform to account for differences in sampling and vertical sensitivity. TROPOMI biases range from +20% at low concentrations ($10^{15}$ molec. cm$^{-2}$) to –13% at higher levels ($15 \times 10^{15}$ molec. cm$^{-2}$). Additionally, we provide seasonal diagnostics, a detailed treatment of uncertainty estimates, and contextualize our findings through a review of recent TROPOMI $NO_2$ validation studies.

## 1 Introduction

Nitrogen oxides ($NO_x = NO + NO_2$) are important trace gases and pollutants in the troposphere. In industrialized areas, they are primarily emitted as NO from fuel combustion associated with anthropogenic activities such as road transport, household heating, power generation, and industry. They also originate from biogenic sources, including bacterial activity in soils and lightning. NO is rapidly converted into $NO_2$ through photochemical reactions which also contribute to the formation of secondary pollutants including tropospheric ozone (Sillman et al., 1990), nitric acid and nitrate aerosols (Chan et al., 2010). $NO_x$ and secondary pollutants all pose threats to human health and the environment (World Health Organization, 2021; European Environment Agency, 2022). They impair respiratory function, particulate matter contributes to cardiovascular diseases, $O_3$ damages crops and vegetation, and $HNO_3$ enhances the eutrophication of water bodies, thereby collectively degrading air and water quality. Moreover, $NO_x$ are reactive species that can exert positive and negative influences on the concentrations of greenhouse gases such as $O_3$ and $CH_4$, and should therefore be incorporated into climate change assessments.



Global coverage of the daily spatial distribution of $NO_2$ is thus a crucial component of atmospheric monitoring. It enables the identification of pollution sources, supports the analysis of spatial and temporal trends, and allows to derive top-down emissions (see, e.g., van der A et al., 2024; Lin et al., 2024). The current state-of-the-art instrument for this purpose is the TROPOspheric Monitoring Instrument (TROPOMI; Veefkind et al. (2012)), a spectrometer-imager onboard the European Space Agency (ESA) polar-orbiting Sentinel-5 Precursor (S-5P) satellite, launched in 2017. TROPOMI follows a series of earlier satellite-borne instruments: the Global Ozone Monitoring Experiment (GOME; Burrows et al. (1999)) launched in 1995, the SCanning Imaging Absorption spectroMeter for Atmospheric CHartographY (SCIAMACHY; Bovensmann et al. (1999)) launched in 2002, and the Ozone Monitoring Instrument (OMI; Levelt et al. (2006); Boersma et al. (2007)) launched in 2004. Across this sequence of instruments, spatial resolution has progressively improved from $40 \times 320$ km$^2$ (GOME), to $30 \times 60$ km$^2$ (SCIAMACHY), $13 \times 24$ km$^2$ (OMI), $3.5 \times 7$ km$^2$ for TROPOMI at its initial resolution, and $3.5 \times 5.5$ km$^2$ since August 2019.

Despite their high relevance, TROPOMI products are subject to limitations and uncertainties arising from the influence of clouds, aerosols, and surface reflection properties on the light path, as well as from uncertainties in the characterization of the a priori vertical profiles of relevant chemical species. Consequently, TROPOMI measurements must be validated against independent observations, preferably with higher spatial and temporal resolution. For instance, the latest Quarterly Validation Report of the S-5P Operational Data Products (Lambert et al., 2025) presents direct comparisons with remote sensing MAX-DOAS instruments globally, showing a positive bias over clean areas ($9.5\%$ for columns below $2 \times 10^{15}$ molec. cm$^{-2}$) and a negative bias over highly polluted areas ($-38\%$ for columns above $15 \times 10^{15}$ molec. cm$^{-2}$). The overall median bias is $-29.4\%$, but it can be reduced by about 20% by smoothing the MAX-DOAS vertical profiles using TROPOMI averaging kernels.

The Small Whiskbroom Imager for atmospheric compositioN monitorinG (SWING) is another type of remote sensing instrument developed at BIRA-IASB to measure tropospheric $NO_2$ from an aircraft and map its distribution over urban areas with high spatial resolution (Merlaud et al., 2018). An upgraded version, SWING+, was developed and deployed during an airborne measurement campaign over Bucharest, the capital city of Romania, comprising 17 flights conducted in 2021 and 2022. Bucharest concentrates significant anthropogenic activity and represents a relatively understudied environment compared to other polluted cities in Europe. In situ measurements within the city consistently exceeded the World Health Organization guideline annual mean limit of 10 $\mu$g m$^{-3}$ for $NO_2$ (World Health Organization, 2021), by up to a factor of 2 in urban areas and up to a factor of 4 near traffic sites in 2021 and 2022 (Ilie et al., 2023). At the same time, Bucharest is surrounded by predominantly rural areas, resulting in sharp spatial gradients in $NO_2$ concentrations due to its short atmospheric lifetime (a few hours in urban settings). SWING+ measurements are acquired at a high spatial resolution of $0.35 \times 0.35$ km$^2$, making them ideal datasets to resolve the plumes emanating from the city and to evaluate TROPOMI products over the Bucharest area. Moreover, the 17 flights span different seasons, allowing for the analysis of seasonal effects, with higher concentrations expected during colder months and lower concentrations during warmer months. A caveat is that SWING+ and TROPOMI acquire measurements with differing vertical sensitivities and at different acquisition times, potentially introducing representation errors in their direct comparison.



In parallel with the measurements, chemical transport models (CTM) provide complementary information on tropospheric
chemical levels. They generate three-dimensional chemical concentration fields at selected time steps based on state-of-the-art theoretical knowledge of atmospheric physics and chemistry, thereby filling the spatial or temporal gaps of observational datasets. In this study, the regional Weather Research and Forecasting model coupled with Chemistry version 4.5.1 (WRF-Chem, Grell et al., 2005; Skamarock et al., 2019) is employed to simulate the atmospheric composition around Bucharest with two nested domains. We use resolutions of $1 \times 1$ km$^2$ over a domain of $100 \times 100$ km$^2$ centered on Bucharest, and $5 \times 5$ km$^2$ over a domain of $400 \times 600$ km$^2$ extending mostly over Romania and Bulgaria. We assess the model predictions through comparisons with in situ meteorological and chemical concentration measurements, as well as with airborne tropospheric column measurements of NO$_2$ from SWING+. Our simulations use the CAMS-REG version 7.0 anthropogenic emission dataset (Kuenen et al., 2022), with an adjustment to NO$_x$ emissions over the city of Bucharest to improve consistency with observations.

Additionally, the use of a CTM such as WRF-Chem enables a more precise comparison between SWING+ and TROPOMI products. By using the model as an intercomparison platform, we can bridge the time lag and account for the respective vertical sensitivities of both instruments, using their averaging kernels. This method was applied by Zhu et al. (2016, 2020) for HCHO over the Southern United States and California, Poraicu et al. (2023) for NO$_2$ over the Antwerp region in Belgium, and is revisited in the present study. Specifically, we exploit the large number of flight measurement days and introduce an explicit treatment of error propagation within this method.

The paper is organized as follows. In Sect. 2, we present the methodology. We begin by briefly describing the WRF-Chem model, including its parameterizations and the selected datasets for boundary and initial conditions, as well as anthropogenic emissions. We then review the measurement datasets used in this study: in situ data for meteorological variables and surface chemical concentrations, and airborne and satellite-borne NO$_2$ tropospheric columns. For the latter two, we detail how WRF-Chem outputs are combined with the instruments averaging kernels to account for their vertical sensitivity. In Sect. 3, we present the results of our analysis. WRF-Chem surface outputs are first evaluated against in situ measurements, with the analysis performed both by combining all available data and by season. Next, the modeled NO$_2$ tropospheric columns are evaluated against SWING+ measurements on a day-by-day basis and by season. TROPOMI columns are then validated against the airborne SWING+ data, using WRF-Chem as an intercomparison platform, with comparisons made both by assembling the full set of flight days and seasonally. In Sect. 4, we review previous validation studies of TROPOMI tropospheric NO$_2$ products and compare them with our own results. Finally, we conclude and summarize our findings in Sect. 5.



## 2   Methodology

### 2.1   The WRF-Chem model

#### 2.1.1   Domain and model setup

We employ the Weather Research and Forecasting model coupled with Chemistry (WRF-Chem) version 4.5.1, along with the
WRF Pre-processing System (WPS) version 4.5 (Grell et al., 2005; Skamarock et al., 2019). Our simulations use two nested
domains centered on Bucharest, Romania. The outer domain covers $400 \times 600$ km$^2$ at a $5 \times 5$ km$^2$ resolution, extending across
Romania and Bulgaria, and also covering parts of the Black Sea, Serbia, Moldova, and Ukraine. The inner domain spans
$100 \times 100$ km$^2$ at a $1 \times 1$ km$^2$ resolution, with its southern and eastern borders intersecting the border between Romania and

Bulgaria (Fig. 1). The vertical grid of the model comprises 44 levels, reaching altitudes up to above 20 km.

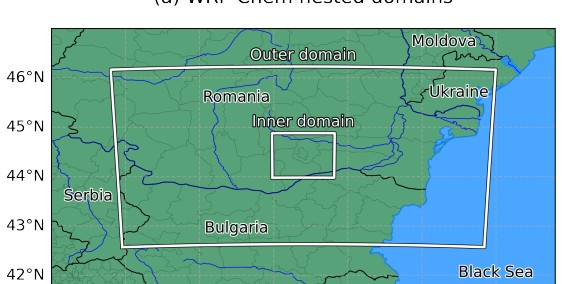

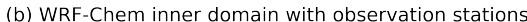

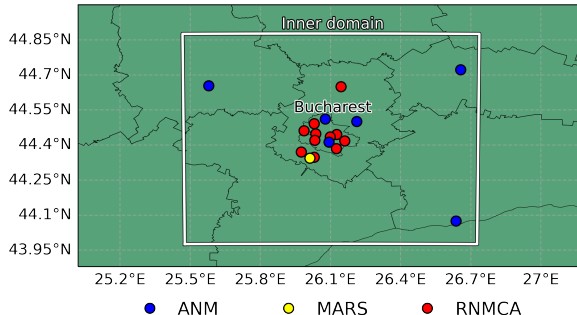

**Figure 1.** (a) WRF-Chem nested domains used for our simulations and (b) closeup of the inner domain showing the municipal borders and
the in situ measurement stations: ANM (blue dots), MARS (yellow) and RNMCA (red). Details are provided in the text.

For each of the 17 SWING+ flights, we ran a WRF-Chem simulation spanning 54 hours, starting at 18:00 UTC two days
before the flight day and ending at 00:00 UTC the day after. This setup allows for comparisons with in situ measurements over



**Table 1.** Summary of the selected physics and chemistry schemes and options used in the WRF-Chem simulations.

|  | Option | Name | Reference(s) |
|---|---|---|---|
| Physics | Cumulus parametrization | Grell-Freitas (GF) | Arakawa (2004); Grell and Freitas (2014) |
|  | Microphysics | Morrison double-moment | Morrison et al. (2009) |
|  | Longwave and shortwave radiation | RRTMG | Iacono et al. (2008) |
|  | Planetary boundary layer scheme | Yonsei University (YSU) | Hong et al. (2006) |
|  | Surface layer scheme | Revised MM5 | Fairall et al. (2003); Jiménez et al. (2012) |
|  | Land surface model | 5-layer thermal diffusion (SLAB) | Dudhia (1996) |
| Chemistry | Gas-phase chemistry | MOZART-4 | Emmons et al. (2010) |
|  | Aerosol chemistry | GOCART | Chin et al. (2000) |

a two-day period (including the day preceding the flight and the flight day itself), with a spin-up time of at least 3 hours since Bucharest operates in UTC+3 or UTC+2 depending on daylight saving time.

The physics and chemistry schemes and options selected for our simulations are summarized in Table 1. In addition to these choices, external data were used. More specifically, static geographical data were obtained at the highest resolution available from the WRF users' webpage (https://www2.mmm.ucar.edu/wrf/users/download/get_sources_wps_geog.html, last access: 22 July 2025). Furthermore, we used the $0.25° \times 0.25°$ ERA5 reanalysis data from ECMWF (Hersbach et al., 2023a, b) to provide the boundary and initial conditions for the physical parameters. These two datasets were regridded to match our nested domains

using the WPS. Boundary and initial conditions for the chemical species are obtained from the $0.95° \times 1.25°$ WACCM6 dataset (Gettelman et al., 2019) and regridded using the mozbc preprocessor, available at the WRF-Chem Tools for the Community webpage (https://www2.acom.ucar.edu/wrf-chem/wrf-chem-tools-community, last access: 22 July 2025).

### 2.1.2   Emissions

We use the CAMS-REG inventory version 7.0 for anthropogenic emissions across the entire domain, with a spatial resolution

of $0.05° \times 0.1°$ (Kuenen et al., 2022). This inventory provides emission maps for each chemical species and for each Gridded Nomenclature For Reporting (GNFR) sector, covering the years 2021 and 2022. It also includes additional temporal factors (hourly, daily, and monthly) and vertical profiles specific to each sector. For emissions from the GNFR sector L, which pertains to agriculture unrelated to livestock, the monthly factors vary by species. The spatial distribution of nitrogen oxides ($NO_x = NO + NO_2$) emission rates, summed across all GNFR sectors and averaged over the years 2021 and 2022, is shown in Fig. 2 (a).

The distribution of $NO_x$ emissions among the GNFR sectors over Bucharest is presented in Fig. 3, along with a comprehensive list of the GNFR sectors. Here and thereafter, we approximate the Bucharest area using the bounding box shown in Fig. 2, defined by $44.34° - 44.53°$ N and $25.96° - 26.24°$ E. Within this area, the yearly anthropogenic $NO_x$ emissions are estimated at 32.6 mol km$^{-2}$ h$^{-1}$ in 2021 and 33.6 mol km$^{-2}$ h$^{-1}$ in 2022. These correspond to $4.03$ and $4.16$ kT y$^{-1}$ of NO emissions, respectively.




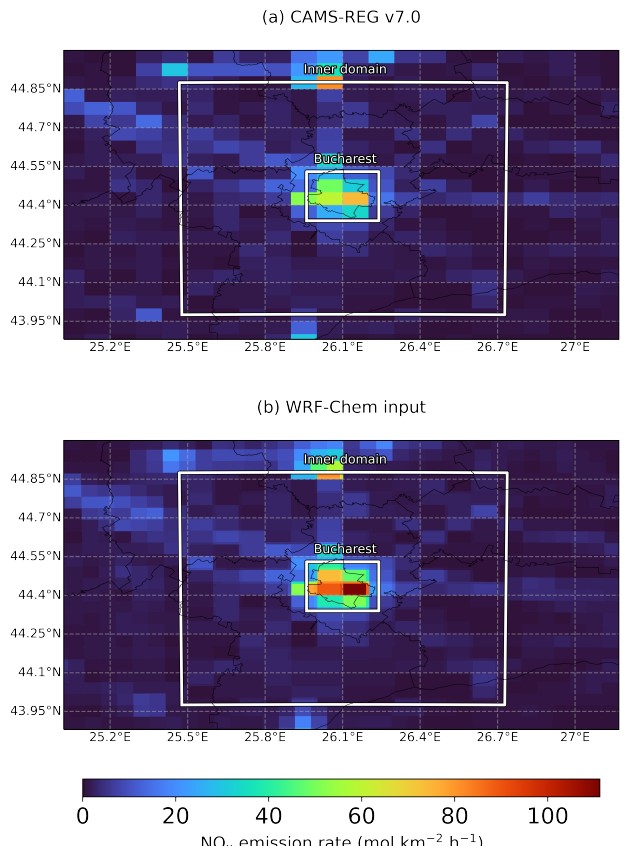

**Figure 2.** Distribution of $NO_x$ emission rates over the WRF-Chem inner domain, summed across all GNFR sectors and averaged for 2021 and 2022. (a) From the CAMS-REG v7.0 inventory at its native resolution. (b) From the CAMS-REG v7.0 inventory, with the emission factor increased by a factor of 1.5 over the Bucharest box, mapped to the WRF-Chem resolution.

Preliminary evaluation using in situ surface concentrations and airborne column measurements indicated that WRF-Chem $NO_2$ levels are generally too low over Bucharest when using CAMS-REG emissions. Therefore, we applied a custom adjustment to the CAMS-REG inventory by multiplying the $NO_x$ emissions by a factor of 1.5 within the previously defined Bucharest box. This brings the yearly fluxes to 48.9 mol km$^{-2}$ h$^{-1}$ in 2021 and 50.4 mol km$^{-2}$ h$^{-1}$ in 2022 over Bucharest. This crude adjustment was estimated based on the model performance in simulations from the 17 flight dates. Its justification

will be made clear from the model comparisons with in situ and airborne measurements (Sect. 3.1.2 and 3.2). We handled the mapping of emissions to match WRF grid cells with a redistribution of the emission mass according to the surface fraction of each WRF grid cell within the corresponding CAMS-REG pixels, preserving the total emitted mass. The resulting map of $NO_x$ emission rates, incorporating the adjustment for Bucharest, as provided to WRF-Chem is presented in Fig. 2 (b). Table 2 provides the mapping of emissions from the CAMS-REG v7.0 inventory to MOZART-4 chemical species. The CAMS-REG





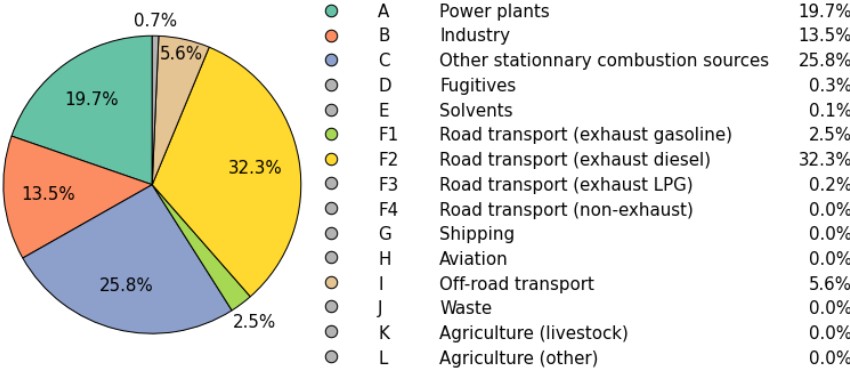

**Figure 3.** Sectoral distribution of anthropogenic $NO_x$ emissions from CAMS-REG v7.0 over the Bucharest box by GNFR sector for the years 2021 and 2022, including a comprehensive list of GNFR sector category codes and names.

volatile organic compounds VOC1, VOC19, and VOC23 are distributed following Chen et al. (2020) (with mass fractions adapted to molar fractions).

Biogenic emissions are computed online by WRF-Chem using the Model of Emissions of Gases and Aerosols from Nature (MEGAN) version 2.04 (Guenther et al., 2006). WRF-Chem input files for the biogenic emissions were generated using the bioemiss preprocessor, available on the WRF-Chem Tools for the Community webpage (https://www2.acom.ucar.edu/

wrf-chem/wrf-chem-tools-community, last access: 22 July 2025).

Lightning-$NO_x$ emissions are computed online based on the parametrization of Price and Rind (1992), which distributes flashes based on convective cloud top height.

## 2.2 Measurements

### 2.2.1 In situ meteorological measurements

The Măgurele center for Atmosphere and Radiation Studies (MARS), located within the WRF inner domain (yellow pin in Fig. 1 (b)), provides measurements of air pressure, temperature, relative humidity, and solar radiation at 2 meters every minute (Carstea et al., 2025). More specifically, the first three aforementioned variables are available only for the first 15 SWING+ dates, while radiation is measured for all of them. When available, these data enable the model evaluation over two-day time series for each SWING+ flight, starting at 00:00 LT on the day preceding the flight and ending at 00:00 LT on the day after.

The national meteorological administration, Administraţia Naţională de Meteorologie (ANM), also called MeteoRomania, provides hourly measurements of air pressure, temperature, relative humidity, solar radiation at 2 meters, and wind speed at 10 meters (https://www.meteoromania.ro/, data acquired upon request on March 13, 2024). The network operates 6 stations located within our WRF inner domain (blue pins in Fig. 1 (b)), named after their respective localities: Afumați, Băneasa, Filaret, Olteniţa, Titu, and Urziceni. For each meteorological variable, we obtained 21 or 22 measurements per flight day and





**Table 2.** Mapping of molar emissions from the CAMS-REG v7.0 inventory to the MOZART-4 mechanism, along with details on labels from both datasets that differ from chemical formulas and standard acronyms or abbreviations.

| MOZART-4 | | CAMS-REG v7.0 | Labeled chemical species | |
|---|---|---|---|---|
| $CH_4$ | ← | $CH_4$ | BIGALK | Alkanes with 4 or more C atoms |
| CO | ← | CO | BIGENE | Alkenes with 4 or more C atoms |
| $NH_3$ | ← | $NH_3$ | TOLUENE | Aromatics |
| NO | ← | $0.90\,NO_x$ | VOC1 | Alcohols |
| $NO_2$ | ← | $0.10\,NO_x$ | VOC2 | Ethane |
| $SO_2$ | ← | $SO_x$ | VOC3 | Propane |
| $CH_3OH$ | ← | $0.26\,(VOC1 + VOC19)$ | VOC4 | Butanes |
| $C_2H_5OH$ | ← | $0.74\,(VOC1 + VOC19)$ | VOC5 | Pentanes |
| $C_2H_6$ | ← | VOC2 | VOC6 | Hexanes and higher alkanes |
| $C_3H_8$ | ← | VOC3 | VOC7 | Ethene |
| BIGALK | ← | $VOC4 + VOC5 + VOC6$ | VOC8 | Propene |
| $C_2H_4$ | ← | VOC7 | VOC9 | Ethyne |
| $C_3H_6$ | ← | VOC8 | VOC10 | Isoprene |
| $C_2H_2$ | ← | VOC9 | VOC11 | Monoterpenes |
| ISOP | ← | VOC10 | VOC12 | Other alk(adi)enes and alkynes |
| $C_{10}H_{16}$ | ← | VOC11 | VOC13 | Benzene |
| BIGENE | ← | VOC12 | VOC14 | Toluene |
| TOLUENE | ← | $VOC13 + VOC14 + VOC15 + VOC16 + VOC17$ | VOC15 | Xylene |
| $CH_2O$ | ← | VOC21 | VOC16 | Trimethylbenzenes |
| $CH_3CHO$ | ← | VOC22 | VOC17 | Other aromatics |
| MEK | ← | $0.35\,VOC23$ | VOC19 | Ethers |
| $CH_3COCH_3$ | ← | $0.65\,VOC23$ | VOC21 | Methanal |
| | | | VOC22 | Other alkanals |
| | | | VOC23 | Ketones |

at each station, with the exception of solar radiation, which was not measured at Titu and was only available for the last 8 SWING+ flight days elsewhere.

### 2.2.2   In situ surface chemical measurements

Surface concentrations of air pollutants are measured hourly by the national air quality monitoring network in Romania, Rețeaua Națională de Monitorizare a Calității Aerului, or RNMCA for short (https://calitateaer.ro/, last access: 22 July 2025).

The network manages 30 monitoring stations in the Bucharest metropolitan area, most of which focus on particulate matter





**Table 3.** List of RNMCA stations in the Bucharest metropolitan area that provide surface concentrations of NO, $NO_2$, and $O_3$. For each species, the number of SWING+ flight overpasses for which the RNMCA station provides in situ measurements is indicated, with 17 being the maximum. The $NO_x/NO_2$ ratio is calculated as the average of the hourly ratios evaluated at night across all two-day measurement series and serves as a criterion for assessing the model representativity at each station.

| RNMCA station | NO | $NO_2$ | $O_3$ | Area type | Nighttime $[NO_x]/[NO_2]$ |
|---|---|---|---|---|---|
| B-1 | 17 | 17 | 17 | urban | 1.47 |
| B-2 | 17 | 17 | 0 | industrial | 1.66 |
| B-3 | 17 | 17 | 0 | traffic | 1.79 |
| B-4 | 17 | 17 | 0 | industrial | 1.45 |
| B-5 | 17 | 17 | 17 | industrial | 1.56 |
| B-6 | 17 | 17 | 0 | traffic | 1.81 |
| B-7 | 17 | 17 | 17 | suburban | 1.49 |
| B-8 | 17 | 17 | 17 | rural | 1.33 |
| B-9 | 7 | 7 | 7 | urban | 1.47 |
| B-10 | 3 | 3 | 0 | urban | 1.64 |
| B-11 | 3 | 3 | 0 | traffic | 1.81 |

measurements. Of these, 11 stations also monitor key chemical species relevant to our study, including NO, $NO_2$, and $O_3$ at some stations. The stations are displayed in Fig. 1 (b). RNMCA provides information about potential pollution sources in the surroundings of each station, allowing the classification into five categories: urban, urban with traffic influence, urban in an industrial area, suburban, and rural, cf. Table 3. For the model evaluation, we consider two-day series of measurements
for each SWING+ overpass. The first data point is recorded at 01:00 LT on the day preceding the flight day, and the last one 47 hours later. This results in 48 data points per flight for each chemical species and each RNMCA station, provided that all measurements are available.

The chemiluminescence measurement of $NO_2$ is known to be affected by interference from compounds in the $NO_y$ reservoir (Lamsal et al., 2008). The modeled mixing ratio of $NO_2$ should therefore account for contributions from PAN, $HNO_3$, and
the sum of alkyl nitrates. The latter includes the (reactive) organic nitrate species, ONIT and ONITR, which are present in the MOZART-4 mechanism. We compute the corrected modeled volume mixing ratio of $NO_2$, referred to as $NO_2^*$, from the WRF-Chem model output as follows:

$$[NO_2^*] = [NO_2] + 0.95[PAN] + 0.35[HNO_3] + [ONIT] + [ONITR]. \tag{1}$$

Hereafter, the measured surface $NO_2$ will be referred to as $NO_2^*$ as well.
Some RNMCA sites are closer to $NO_x$ pollution sources than others and are more likely to show higher $NO_x$ concentrations than the model prediction due to enhanced representation errors. Poraicu et al. (2025) suggested that the measured nighttime $NO_x/NO_2$ ratio can be used to determine whether a station is well represented by the model. Indeed, away from emission





sources, $NO_x$ species are expected to reach the pseudo-steady state (PSS) of their photochemical cycle, which constrains the $NO_x/NO_2$ ratio:


$$\left(\frac{[NO_x]}{[NO_2]}\right)_{PSS} = 1 + \frac{J}{k[O_3] + \ldots}, \tag{2}$$

where $J$ is the photolysis rate of $NO_2$, $k = 1.95 \times 10^{-14}$ molec.$^{-1}$ cm$^3$ s$^{-1}$ (at 298 K) is the rate constant for the titration of $O_3$ with NO (Burkholder et al., 2019), and the dots represent contributions from peroxy radicals. At night, $J$ becomes negligible, causing the ratio to decrease and approach 1. However, photochemical equilibrium is far from being achieved near $NO_x$ pollution sources, which primarily emit NO and lead to observed ratios significantly higher than 1. Thus, we select RNMCA

stations with relatively low measured $NO_x/NO_2$ ratios during nighttime, in order to exclude the least representative stations. As shown in Table 3, stations influenced by traffic exhibit the largest deviations from the PSS prediction, with nighttime ratios greater or equal to 1.79. As expected, the lowest ratio (1.33) is found for the only rural station. For the model evaluation in Sect. 3.1.2, we will focus on the eight stations not directly exposed to traffic, characterized by a nighttime $NO_x/NO_2$ ratio below 1.7 (B-1, B-2, B-4, B-5, B-7, B-8, B-9, and B-10). The enhancement of $NO_2$ concentrations at traffic stations is not

specific to our selected dates but was also observed in yearly averages from 2020 to 2022, as reported by Ilie et al. (2023). Note that this distinction does not affect the analysis of $O_3$ concentrations, as traffic stations do not measure it. Ozone is only monitored at five distinct stations (Table 3).

### 2.2.3 Airborne SWING+ NO₂ column measurements

SWING (Small Whiskbroom Imager for atmospheric compositioN monitorinG) instruments are compact whiskbroom imagers

developed at BIRA-IASB for air quality mapping. They use ultraviolet and visible-light spectrometers, covering a spectral range of $280 - 550$ nm with a resolution of 0.7 nm Full Width Half Maximum (FWHM), to retrieve $NO_2$ column abundances using the Differential Optical Absorption Spectroscopy (DOAS) technique (Platt and Stutz, 2008). Initially designed for operations onboard an unmanned aerial vehicle (UAV) (Merlaud et al., 2018), SWING instruments have since been deployed on crewed aircraft for validation flights alongside larger airborne imagers over Berlin (Tack et al., 2019) and Bucharest (Merlaud

et al., 2020).

The SWING observations over Bucharest exploited in this study originate from two ESA-funded projects: RAMOS (Nemuc et al., 2023) and QA4EO (Nemuc et al., 2024). Within RAMOS, a custom version of the instrument, named SWING+, was developed at BIRA-IASB and permanently installed on the Britten-Norman 2 (BN-2) aircraft operated by INCAS (National Institute for Aerospace Research). Compared to the original UAV version, SWING+ is enclosed in an aluminum casing, with

the scanner deported by 20 cm to exit the aircraft fuselage. The instrument is still relatively compact ($45 \times 19 \times 15$ cm$^3$, 3.8 kg).

In contrast to typical field campaigns that are typically deployed over a few weeks during summer, the flight strategy in RAMOS and QA4EO consisted in flying on a regulatory basis across the year, limited to clear-sky conditions. The BN-2 hovered the city from an altitude of 3 km, and the SWING+ swath was set to $48°$, incremented in steps of $6°$, with an integration time of with 0.5 s. This configuration resulted in a ground resolution of $0.35 \times 0.35$ km$^2$. Table 4 lists the 17 flights



**Table 4.** Acquisition start and end times (local time) of the SWING+ and TROPOMI instruments for each flight date (dd/mm/yyyy) over Bucharest.

| Dates | SWING+ start | SWING+ end | TROPOMI start | TROPOMI end |
|---|---|---|---|---|
| 01/07/2021 | 10:31:17 | 12:05:13 | 14:13:09 | 14:13:12 |
| 05/07/2021 | 13:28:27 | 15:11:13 | 14:38:20 | 14:38:24 |
| 10/07/2021 | 12:47:44 | 14:05:20 | 13:04:56 | 13:05:00 |
| 29/10/2021 | 12:59:26 | 14:13:12 | 13:24:05 | 13:24:09 |
| 04/11/2021 | 11:35:57 | 13:02:46 | 12:11:47 | 12:11:52 |
| 05/11/2021 | 12:21:00 | 14:02:01 | 13:32:45 | 13:32:48 |
| 11/11/2021 | 12:00:07 | 13:54:36 | 13:20:12 | 13:20:17 |
| 22/11/2021 | 12:04:26 | 14:01:39 | 13:14:00 | 13:14:05 |
| 23/12/2021 | 12:06:58 | 14:16:26 | 13:33:06 | 13:33:11 |
| 05/01/2022 | 11:38:25 | 13:41:06 | 12:49:21 | 12:49:25 |
| 24/03/2022 | 12:16:01 | 14:15:00 | 13:26:56 | 13:27:01 |
| 28/03/2022 | 12:26:05 | 14:03:22 | 13:12:15 | 13:12:18 |
| 05/04/2022 | 12:48:17 | 14:51:50 | 14:01:47 | 14:01:52 |
| 15/04/2022 | 13:16:21 | 15:11:11 | 14:14:13 | 14:14:18 |
| 30/06/2022 | 12:55:39 | 14:25:13 | 13:48:38 | 13:48:42 |
| 30/09/2022 | 12:57:40 | 14:39:19 | 13:24:07 | 13:24:13 |
| 02/11/2022 | 11:24:43 | 12:46:42 | 12:05:54 | 12:05:57 |

used in this study, operated between July 2021 and November 2022. All dates are weekdays, except for 10/07/2021, which was a Saturday. Flight times were chosen to coincide with TROPOMI overpasses, except for the first date which was the test flight.

In the DOAS analysis, each vertical column density (VCD, or $\Omega_S$ when specifically referring to SWING+) of $NO_2$ is obtained by dividing the slant column density (SCD) by an air mass factor (AMF) specific to each measurement. The slant column itself is the sum of a reference slant column density ($SCD_{ref}$), estimated only once per flight, and the differential slant column density

(DSCD):

$$\text{VCD} = \frac{\text{SCD}}{\text{AMF}} = \frac{\text{SCD}_{ref} + \text{DSCD}}{\text{AMF}}. \tag{3}$$

AMFs are computed using the uvspec/DISORT radiative transfer model (Mayer and Kylling, 2005), with a relative uncertainty of $15.2\%$ across the dataset. $SCD_{ref}$ values are daily averages derived from 30 reference spectra over a clean area, with an uncertainty of $0.5 - 2.1 \times 10^{15}$ molec. $cm^{-2}$, propagating as a $0.2 - 1.0 \times 10^{15}$ molec. $cm^{-2}$ error on the VCD. Averaged

per flight day, DSCD uncertainty ranges from $1.4 - 2.5 \times 10^{15}$ molec. $cm^{-2}$, reducing to $0.5 - 1.6 \times 10^{15}$ molec. $cm^{-2}$ when propagated into the VCD. Summed in quadrature, these errors yield a total VCD uncertainty of $0.9 - 1.9 \times 10^{15}$ molec. $cm^{-2}$. Lower uncertainties correspond to lower VCDs observed in spring and summer, while higher uncertainties are associated with elevated columns in fall and winter.





For the evaluation of the WRF-Chem model, SWING+ vertical column densities and averaging kernels are regridded to the
model resolution. Measurements falling within the same WRF grid cell and separated in time by less than the model output
interval (5 minutes) are averaged to produce a single regridded SWING+ column. After regridding, the daily average VCD
error due to DSCD uncertainty, which is primarily of random origin, decreases to $0.3 - 0.7 \times 10^{15}$ molec. cm$^{-2}$. Systematic
errors remain unaffected by the regridding. The same process is applied to the averaging kernels, denoted as $A_\mathrm{S}$, which are
then used to evaluate the modeled columns, accounting for the instrument vertical sensitivity. More precisely, WRF-Chem
NO$_2$ tropospheric columns $\Omega_\mathrm{W,S}$ are derived from the modeled NO$_2$ density field $n_\mathrm{W}$ and the regridded kernels by integrating
over the troposphere (Trop):

$$\Omega_\mathrm{W,S} = \int\limits_\mathrm{Trop} A_\mathrm{S}(z) n_\mathrm{W}(z) \mathrm{d}z\,. \tag{4}$$

The regridding of SWING+ measurements for the purpose of TROPOMI validation is detailed in the next section.

### 2.2.4 Satellite-borne TROPOMI NO$_2$ column measurements

The TROPOspheric Monitoring Instrument (TROPOMI) was launched aboard the Sentinel-5 Precursor (S5P) satellite of the
European Space Agency in October 2017 to monitor atmospheric composition and air quality (Veefkind et al., 2012). S5P
is a near-polar and sun-synchronous satellite with a near-daily overpass. TROPOMI is a nadir-viewing pushbroom imaging
spectrometer that covers spectral bands in the ultraviolet, visible, near-infrared, and shortwave infrared regions, enabling the
retrieval of key atmospheric trace gases, including NO$_2$. Its spatial resolution was initially $7 \times 3.5$ km$^2$, and improved to
$5.5 \times 3.5$ km$^2$ after August 2019. Its overpass times over Bucharest are listed in Table 4.

The TROPOMI tropospheric NO$_2$ vertical column density $\Omega_\mathrm{T}$ is generated through a multi-step retrieval process. First,
differential optical absorption spectroscopy (DOAS) is applied to the Level-1b radiance and irradiance spectra to retrieve total
slant column densities in the $405 - 465$ nm range, using techniques developed for OMI (van Geffen et al., 2020). Second,
the separation of stratospheric and tropospheric contributions is performed using data assimilation in the TM5-MP chemistry
transport model (Williams et al., 2017). In the final step, the tropospheric slant column is converted to a vertical column using
air mass factors (AMFs), which are computed with the Doubling-Adding KNMI radiative transfer model (de Haan et al., 1987;
Stammes, 2001) based on TM5-MP NO$_2$ vertical profiles. Further details are provided in the TROPOMI NO$_2$ Algorithm
Theoretical Basis Document (van Geffen et al., 2024).

In this study, we evaluate TROPOMI NO$_2$ retrievals from version 2.4.0, using reprocessed data (RPRO) up to 17/07/2022 and
offline products (OFFL) thereafter (Eskes et al., 2024). This version incorporates an updated surface albedo climatology based
on TROPOMI observations. Only measurements with a quality assurance value greater than 0.75 are retained, in accordance
with the recommendation. Additionally, only those TROPOMI measurements for which at least 50% of the pixel area is
covered by SWING+ observations are considered in the analysis. The average precision of these TROPOMI measurements is
$1.3 \times 10^{15}$ molec. cm$^{-2}$.

SWING+ measurements are used here to validate TROPOMI, with WRF-Chem serving as an intercomparison platform that
accounts for the acquisition times and vertical sensitivities of both instruments. The validation is carried out in two steps:





1. We assess the bias of WRF-Chem relative to SWING+ and determine the appropriate correction for its columns for each flight. This is realized at the TROPOMI spatial resolution by averaging both SWING+ and the corresponding WRF-Chem columns $\Omega_{\mathrm{W,S}}$ over TROPOMI pixels. At this resolution, the random uncertainty on the SWING+ column stemming from the DSCD, presented in the previous section, falls below $10^{14}$ molec. cm$^{-2}$.

2. We evaluate another set of WRF-Chem columns, denoted as $\Omega_{\mathrm{W,T}}$, using TROPOMI averaging kernels $A_{\mathrm{T}}$ and the modeled NO$_2$ density profile $n_{\mathrm{W}}$ averaged over TROPOMI pixels:

$$\Omega_{\mathrm{W,T}} = \int\limits_{\mathrm{Trop}} A_{\mathrm{T}}(z) n_{\mathrm{W}}(z) \mathrm{d}z, \tag{5}$$

and correct these columns based on the insights gained from the first step. The bias-corrected version of $\Omega_{\mathrm{W,T}}$ columns, denoted by $\Omega_{\mathrm{W,T}}^{\mathrm{bc}}$, then serve as a reference to evaluate the bias of TROPOMI, combining data from different flight days, either all together or by season.

In both steps, we assume that the biases of the model and the satellite can be captured through linear regression against reference values. To ensure the quality of the results, a selection of flight days will be made based on the evaluation of the model against SWING+ data. Note that this method generalizes the approach of Poraicu et al. (2023) by extending it to simultaneously address multiple flight days. In their study, WRF-Chem biases relative to the airborne instrument APEX and to TROPOMI were subtracted to infer the bias of TROPOMI with respect to APEX, one flight date at a time.

## 3 Results

### 3.1 Model evaluation using in situ measurements

#### 3.1.1 Meteorological observations

In this section, we present the results of the model evaluation for the surface values of physical parameters measured at the MARS and ANM stations. The analysis combines the observed and modeled physical parameters from all 17 flight dates, over the corresponding two-day periods at MARS and the flight days for the ANM stations (see Fig. 4). Details about the synoptic parameters specific to individual SWING+ flight days that could be critical for the evaluation of the NO$_2$ column, such as the modeled wind direction over the city during the flight time, will be discussed in Sect. 3.2. Throughout this study, we use the statistical metrics and abbreviations defined in Table 5.

The model MB for air pressure is 1.0 mbar at MARS, 0.4 mbar at the ANM stations, and overall negligible in terms of relative biases. The corresponding RMSE are 1.2 mbar and 0.9 mbar, respectively. Both measurement datasets show a perfect correlation coefficient of 1.00. The air temperature measurements indicate model biases of $-0.4$°C at MARS and 0.1°C at the ANM stations. The daytime underestimation and nighttime overestimation of temperature were reported in a previous study using the WRF model over Bucharest (Iriza et al., 2017). The model RMSE reach 2.3 and 2.4°C, respectively. The correlation remains excellent overall, with Pearson's coefficients of 0.98 and 0.97. The MB of the model for relative humidity reach 2.1%





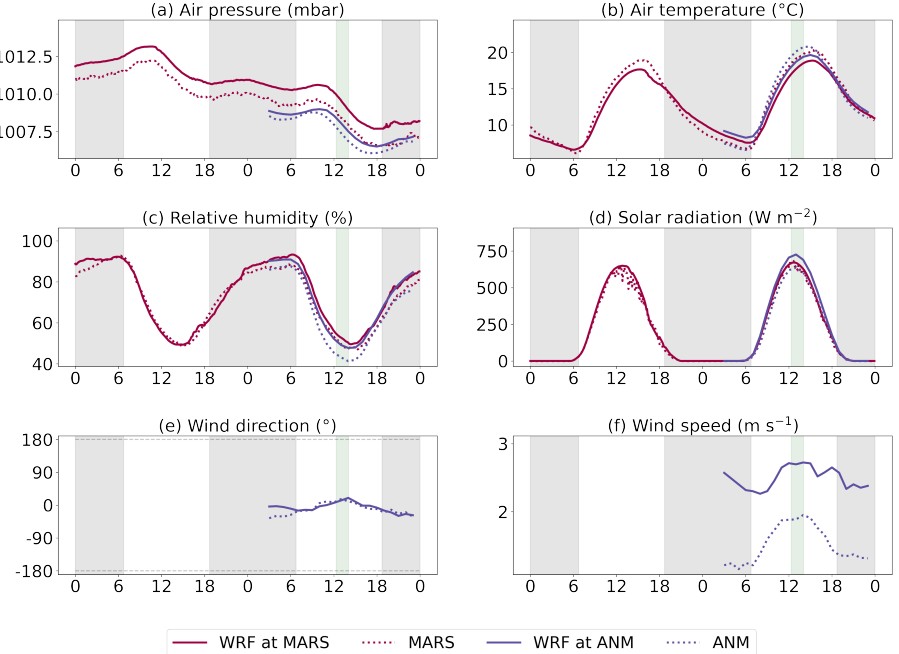

**Figure 4.** Surface meteorological measurements from MARS and ANM and comparison with WRF-Chem. The horizontal axes represent local time in hours. Each plot focuses on a specific meteorological parameter and includes all 17 one or two-day time series from the various stations, when available, averaged into a single time series. Some points were excluded from the plots when the number of available station-date pairs fell below 95% of the maximum, as this was considered unrepresentative visually, though the data is still included in the main text analysis. Gray and green windows indicate the averaged nighttime and SWING+ flight hours from the measurement dates, respectively.

at MARS and 5.4% at the ANM stations. RMSEs are an order of magnitude higher, 11.5% and 14.2%, respectively. High correlations, with values of 0.86 and 0.85, are calculated at the corresponding sites. Solar radiation is well reproduced by the model according to MARS measurements: the MB of 9.2 W m$^{-2}$ is negligible, the RMSE is 66.7 W m$^{-2}$, and the correlation is close to 1 (0.97). Generally, fewer fluctuations are observed on the second day, see Fig. 4 (d), as it was selected as the flight day due to favorable weather conditions. The good model performance is further confirmed with the ANM measurements, despite an increase in bias and error: the MB is 36.5 W m$^{-2}$, the RMSE is 69.8 Wm$^{-2}$, and the correlation coefficient is equal to 0.99. The ANM measurements indicate an overestimation of the modeled wind speed by 1.0 m s$^{-1}$, in line with former WRF evaluations over urban areas (Kim et al., 2013; Feng et al., 2016; Poraicu et al., 2023). The wind direction is biased by 15.7°. Evaluating both components of the modeled horizontal wind field, $U$ and $V$, the RMSE is 1.5 m s$^{-1}$ and we find a Pearson's correlation coefficient of 0.64.





**Table 5.** Statistical metrics used to evaluate the model. The formulas are written for $N$ observed values $O_i$ and the corresponding modeled data $M_i$, with $i = 1, \ldots, N$.

| Metric | Formula |
|---|---|
| Mean observed value | $\overline{O} = \dfrac{1}{N} \sum_{i=1}^{N} O_i$ |
| Mean modeled value | $\overline{M} = \dfrac{1}{N} \sum_{i=1}^{N} M_i$ |
| Mean bias | $\text{MB} = \overline{M} - \overline{O}$ |
| Relative bias | $\text{RB} = \dfrac{\overline{M} - \overline{O}}{|\overline{O}|}$ |
| Root mean square error | $\text{RMSE} = \sqrt{\frac{1}{N} \sum_{i=1}^{N} (M_i - O_i)^2}$ |
| Pearson's correlation coefficient | $r = \dfrac{\sum_{i=1}^{N} (O_i - \overline{O})(M_i - \overline{M})}{\sqrt{\sum_{j=1}^{N} (O_j - \overline{O})^2} \sqrt{\sum_{k=1}^{N} (M_k - \overline{M})^2}}$ |

### 3.1.2 Surface chemical concentrations

We compare surface concentration measurements from the RNMCA network with WRF-Chem for NO, $NO_2^*$, and $O_3$ at the model lowest vertical level. Figure 5 displays the model performance for the different types of stations. Each plot includes

measurements from the 17 two-day time series, averaged into a single time series. Therefore, the discussion of seasonality is deferred to a later part of this section.

The model generally underestimates NO and $NO_2^*$ levels while overestimating $O_3$, in line with the strong titration effect of $NO_x$ on ozone near pollution sources. The underestimation of $NO_2^*$ is significantly reduced when moving from traffic stations ($-21$ $\mu$g m$^{-3}$) to suburban and rural stations ($-5$ $\mu$g m$^{-3}$). This aligns with the discussion on station representativeness in

Sect. 2.2.2 and further supports the exclusion of traffic stations when selecting representative sites. We therefore present in Table 6 the statistical metrics used to evaluate the 17 two-day time series, focusing only on non-traffic stations.

The negative bias in daytime $NO_2$ levels in WRF-Chem is similar to the reported underestimation by Poraicu et al. (2023). However, WRF-Chem simulations over Europe have shown an important nighttime overestimation of $NO_2$ (Poraicu et al., 2023; Kuhn et al., 2024), which is not observed here. Our results also contrast with those from the Land-Use Regression

model (Talianu et al., 2024), which reported daytime positive biases of $8-30\%$ during a period within 2022 and 2023, using a comparable set of measurements over Bucharest.

Several factors may explain this discrepancy. While an underestimation of emissions remains a possibility, comparisons with $NO_2$ column measurements in the following sections suggest that the factor of 1.5 applied to the CAMS-REG inventory is well-justified. However, other factors could also contribute to the model underestimation:

– Poor representativeness of measurements: Even at non-traffic stations, nighttime $NO_x/NO_2$ ratios remain significantly higher than 1 (see Table 3).



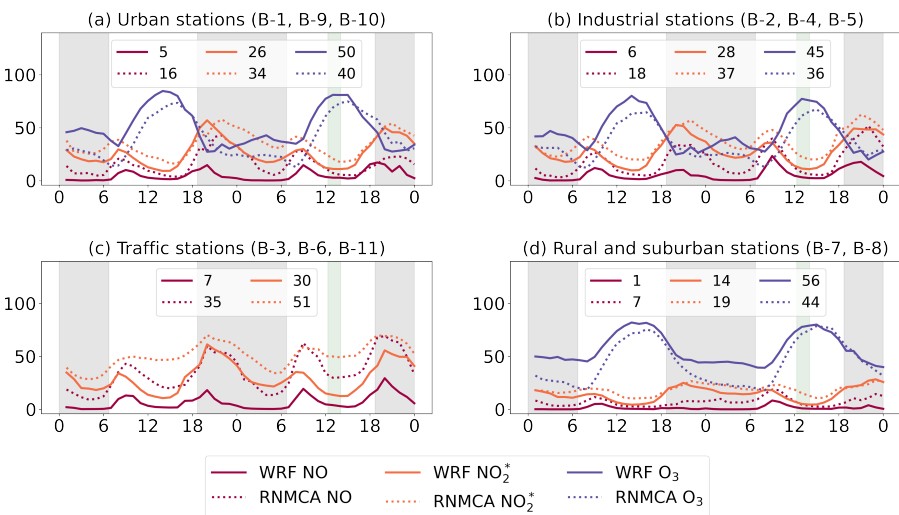

**Figure 5.** Comparison of surface NO, $NO_2^*$, and $O_3$ measurements from the RNMCA network with WRF-Chem model outputs. Units of the vertical and horizontal axes are $\mu g\ m^{-3}$ and hours (local time). Each plot focuses on a specific set of RNMCA stations based on location and includes all 17 two-day time series, when available, averaged into a single time series. For each series, we indicate the mean value in the legend. Gray and green windows indicate the averaged nighttime and SWING+ flight hours from the measurement dates, respectively.

  – Limited resolution of anthropogenic emissions: The CAMS-REG inventory is too coarse to accurately capture the spatial heterogeneity of the city, leading to an underestimation of $NO_x$ pollution levels near hotspots.

  – Overestimated surface wind: As seen in Sect. 3.1.1, the model overestimates horizontal wind speed, which enhances the advection of clean air from surrounding rural areas, diluting $NO_x$ concentrations.

  – Excessive vertical mixing: Turbulence in the boundary layer could further contribute to the dilution of $NO_x$ species. Unfortunately, the lack of ceilometer data prevents us from diagnosing potentially overestimated vertical mixing.

The model generally captures well the diurnal cycle of the measurements. The rush hour peak in the morning and the evening peak in $NO_x$ are observed in both the measured and modeled concentrations. As pointed out by Poraicu et al. (2023), a rush hour peak in the late afternoon is not always identifiable, which is expected due to the counterbalancing effects of chemical sink and boundary layer development. The daytime ozone buildup and plateau are well reproduced by the model. Slight delays in these patterns may occur, but the overall correlation is satisfactory. During daytime, we report in Table 6 correlation coefficients of 0.70, 0.71, and 0.81 for NO, $NO_2^*$, and $O_3$, respectively. Notably, if we restrict the time window to flight hours for $NO_2^*$, in anticipation of the SWING+ measurements analysis, the correlation increases to 0.80.

The two-day simulation periods may be grouped according to the meteorological seasons. Figure 6 compares the measured values at non-traffic RNCMA stations with the corresponding modeled outputs.





**Table 6.** Statistical metrics calculated for each species and period of the day. Values are obtained from non-traffic RNMCA stations and flight hours refer to the SWING+ acquisition times.

| Species | Time period | MB ($\mu g\ m^{-3}$) | RB (%) | RMSE ($\mu g\ m^{-3}$) | PCC $r$ |
|---------|-------------|------|------|------|------|
| NO | Two days | −10 | −70 | 28 | 0.48 |
| | Daytime | −7 | −61 | 16 | 0.70 |
| | Nighttime | −14 | −76 | 37 | 0.41 |
| | Flight hours | −4 | −60 | 7 | 0.67 |
| $NO_2^*$ | Two days | −7 | −24 | 20 | 0.66 |
| | Daytime | −8 | −33 | 15 | 0.71 |
| | Nighttime | −6 | −17 | 24 | 0.58 |
| | Flight hours | −9 | −49 | 12 | 0.80 |
| $O_3$ | Two days | 11 | 27 | 24 | 0.76 |
| | Daytime | 12 | 22 | 22 | 0.81 |
| | Nighttime | 10 | 38 | 25 | 0.47 |
| | Flight hours | 9 | 12 | 16 | 0.85 |

NO peaks are sharper during cold months. This is due to lower sun exposure in winter, which reduces the generation of ozone and peroxy radicals, both of which are sinks for NO, thereby increasing its lifetime as well as the time needed to reach photochemical steady state between NO and $NO_2$. As in the previous analysis, we find that NO levels are generally underestimated by the model across all seasons. In particular, the model does not simulate enough nighttime accumulation during the colder months. The second day of the winter runs shows the best agreement during the daytime, but since this analysis is based on only two time series, it is difficult to draw definitive conclusions. Daytime correlation values remain consistent across seasons, ranging from 0.60 in winter to 0.68 in fall. Note that in summer, measurements are close to the detection limit.

Similarly to NO, surface levels of $NO_2^*$ are generally underestimated. The best agreement is found in winter, where nighttime values appear to be particularly well reproduced. The diurnal evolution during this season is also well captured, though with greater variation. The morning rush hour peak is visible in the modeled values for fall and spring but is too flat during summer. Daytime correlation coefficients range from 0.59 in summer to 0.73 in winter.

Ozone is consistently overestimated across all seasons, both during day and night. As expected, months with higher sun exposure exhibit a more significant $O_3$ buildup, generated from the oxidation of carbon monoxide and volatile organic compounds in presence of $NO_x$ and ultraviolet radiation. This seasonal variability is present in both the measured and modeled values (in value ranges comparable to those observed in the WRF-Chem simulations of Maco et al. (2019)). Notably, a good agreement is found at the summer daytime maxima. Daytime correlation is lower in winter, with a coefficient of 0.49, while in other seasons, it ranges from 0.75 to 0.79.





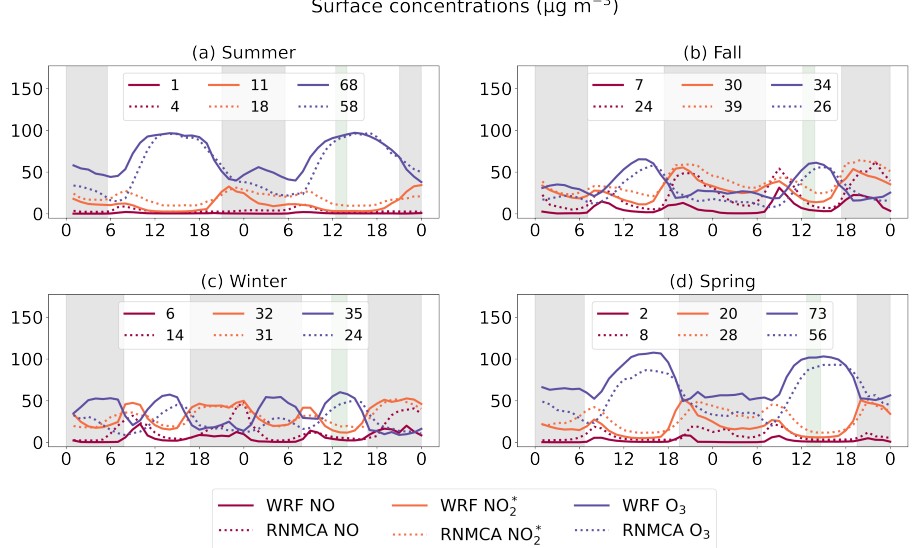

**Figure 6.** Comparison of surface RNMCA measurements from non-traffic stations with WRF-Chem model outputs. Units of the vertical and horizontal axes are $\mu g\,m^{-3}$ and hours (local time). Each plot is the averaged curve of series of two-days on a specific meteorological season: summer (4 series), fall (7 series), winter (2 series), and spring (5 series). For each two-day series, we indicate the mean value of the surface concentration in the legend. Gray and green windows indicate the averaged nighttime and SWING+ flight hours from the measurement dates, respectively.

## 3.2 Model evaluation against airborne column measurements

For each flight, column comparisons are assessed using the statistical metrics of Table 5. The results vary significantly from one date to the next. Therefore, we will first provide a detailed analysis only for the two flight days with the highest correlation coefficient, 11/11/2021 and 30/06/2022, before presenting a summary encompassing all flight days.

### 3.2.1 Flight on November 11, 2021

The temporal series and maps in Fig. 7 illustrate the model capability to reproduce tropospheric $NO_2$ columns on our best-performing date. The observed and modeled $NO_2$ levels are very similar, and the synchronicity of the peaks and dips in the time series leads to an excellent correlation coeffcient of 0.94. The maps clearly show a plume emanating from the city and transporting $NO_2$ in the North-West direction in both cases. This is a satisfactory result considering the coarse resolution of the emission inventory.

The calculated RMSE is equal to $1.7 \times 10^{15}$ molec. $cm^{-2}$, mainly due to an overestimation, both in the background values and at the plume peaks. The RB is relatively high (43%), partly due to the large number of small values, including negative ones, in the SWING+ measurements. Note that negative values may result from a calibration offset combined with random





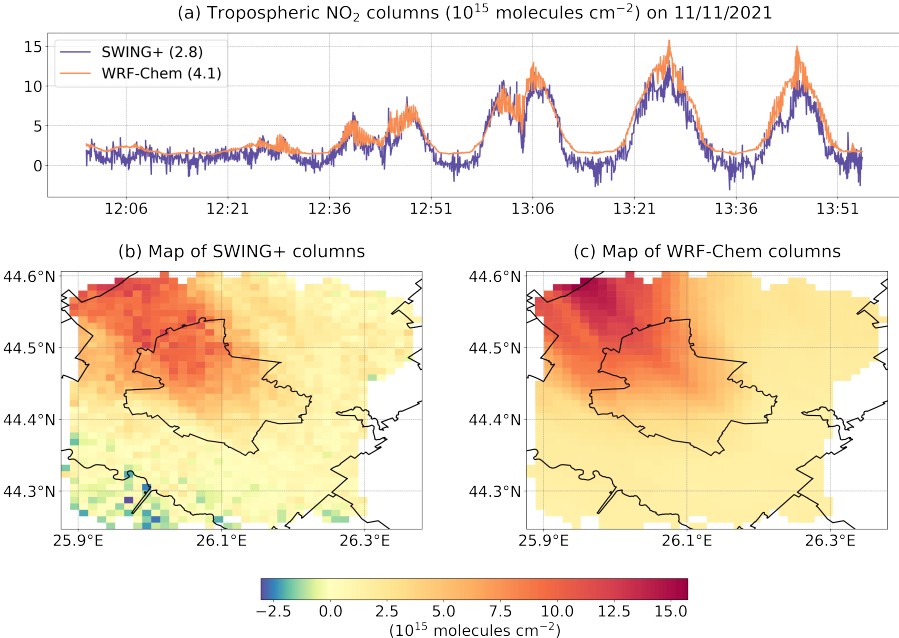

**Figure 7.** Tropospheric $NO_2$ columns on Thursday 11 November 2021, presented as a temporal series of SWING+ and WRF-Chem values plotted against local time, with mean values in parentheses in (a), and corresponding maps in (b) and (c).

errors in the background values. For this flight, only 2.3% of the measurements are negative and within the bounds of the error bar.

### 3.2.2 Flight on June 30, 2022

Figure 8 presents the model evaluation on June 30, 2022. The first part of the airborne measurements shows abnormally high background values away from the plume emanating from the city (cf. dotted area in the subfigures). Those high values are not captured by the model and are likely due to a stabilization delay of the SWING+ instrument. Specifically, since the SWING+ instrument is not thermally stabilized, its spectral resolution changes as the temperature decreases during the flight ascent. These variations affect the accuracy of the $NO_2$ measurements. For this reason, we exclude the first measurements, up to 13:24 LT, from our analysis.

The modeled columns correlate very well with the measurements, with a Pearson's coefficient of 0.89. This time, however, the model tends to underestimate the measurements, with a small RB of $-18\%$ and an error of $1.5 \times 10^{15}$ molec. $cm^{-2}$. This latter metrics is dominated by small columns associated with the background, where the model slightly overestimates the measurements, similarly to what was found for November 11, 2021, though to a lesser extent.





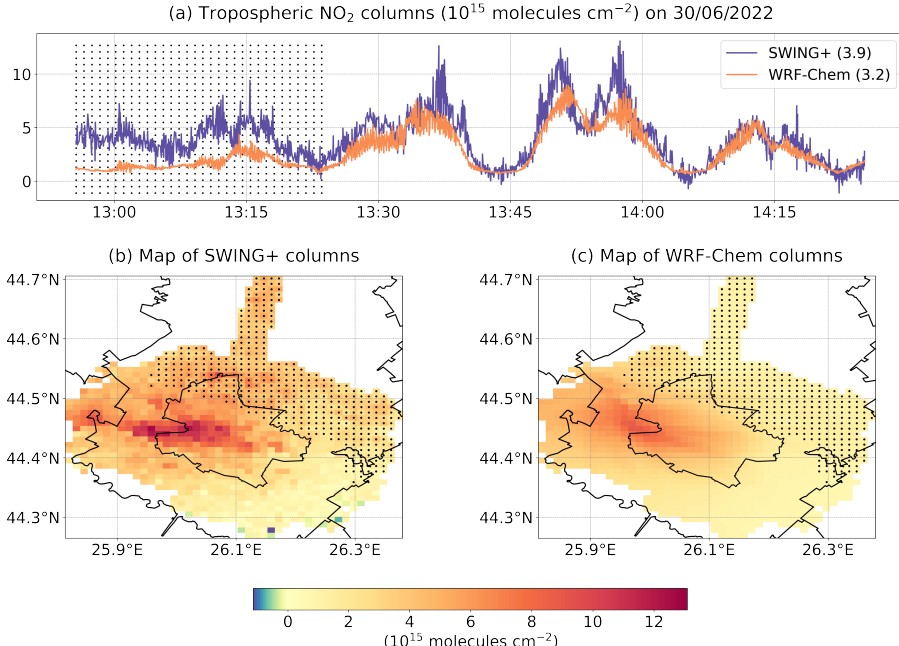

**Figure 8.** Tropospheric NO$_2$ columns on Thursday 30 June 2022, presented as a temporal series of SWING+ and WRF-Chem values plotted against local time, with mean values in parentheses in (a), and corresponding maps in (b) and (c). Dotted values, acquired from 12:55 to 13:24 LT, are excluded from the analysis for reasons explained in the text.

### 3.2.3 Summary for all flights

Table 7 presents the evaluation of NO$_2$ columns from WRF-Chem against SWING+ measurements using the statistical metrics from Table 5, for each separate flight. In addition to 30/06/2022, another flight date required data truncation. Inspection of both datasets, conducted independently of each other, indicated that selecting 13:24 LT as the start time was appropriate. It also

provides statistics per season and for the entire dataset. For two dates, reported in the table, we truncate data associated with the beginning of the flight for reasons explained in Sect. 3.2.2. Equivalents of Fig. 7 and Fig. 8 for the other flight dates are presented in Supplement 1.

The specific case of 22/11/2021 stands out as an outlier due to its large RMSE ($11.2 \times 10^{15}$ molec. cm$^{-2}$) and a correlation coefficient close to 0 ($-0.05$). A detailed inspection of the model meteorological performance for that day, in comparison

with ANM measurements, reveals that it fails to accurately reproduce a change in surface wind direction just before SWING+ begins recording. The observations indicate a transition from westerly to easterly winds occurring between 05:00 and 09:00 LT. In contrast, the model simulates this transition beginning around 08:00 and completing near 13:00, resulting in a delay of approximately three to four hours. This issue justifies the omission of this flight from further analyses.

The comparable numbers of days with either positive (7) or negative (10) biases in Table 7 suggest a balanced model

behavior. Along with the small overall bias (MB of $0.5 \times 10^{15}$ molec. cm$^{-2}$ and RB of 13%) across all selected dates, this



**Table 7.** Evaluation of tropospheric $NO_2$ columns from WRF-Chem against SWING+ measurements, regridded to the resolution of the model, for each flight day. For dates marked with a dagger (†), measurements have been truncated to start at the time of 13:24 LT. The last rows assembles data by season or for all dates combined, excluding the worst-performing one, 22/11/2021, when marked with an asterisk (∗).

| Dates | Sample size | MB ($10^{15}$ molec. cm$^{-2}$) | RB (%) | RMSE ($10^{15}$ molec. cm$^{-2}$) | PCC $r$ |
|---|---|---|---|---|---|
| 01/07/2021 | 1531 | −0.3 | −11 | 0.8 | 0.80 |
| 05/07/2021 | 1436 | 0.2 | 9 | 1.5 | 0.59 |
| 10/07/2021[†] | 677 | −0.1 | −4 | 0.9 | 0.58 |
| 29/10/2021 | 1355 | 4.6 | 79 | 6.5 | 0.86 |
| 04/11/2021 | 1333 | −0.3 | −5 | 2.3 | 0.85 |
| 05/11/2021 | 1691 | 5.3 | 125 | 7.7 | 0.69 |
| 11/11/2021 | 1902 | 1.2 | 43 | 1.7 | 0.94 |
| 22/11/2021 | 1899 | 3.9 | 58 | 11.2 | −0.05 |
| 23/12/2021 | 2020 | 4.6 | 113 | 5.1 | 0.72 |
| 05/01/2022 | 1937 | 0.2 | 7 | 1.1 | 0.79 |
| 24/03/2022 | 1985 | −1.0 | −24 | 2.2 | 0.59 |
| 28/03/2022 | 1617 | −0.5 | −15 | 2.1 | 0.56 |
| 05/04/2022 | 1936 | −0.8 | −34 | 1.3 | 0.75 |
| 15/04/2022 | 2038 | −1.3 | −31 | 2.4 | 0.76 |
| 30/06/2022[†] | 1136 | −0.7 | −18 | 1.5 | 0.89 |
| 30/09/2022 | 1772 | −1.6 | −33 | 3.4 | 0.70 |
| 02/11/2022 | 1590 | −1.6 | −25 | 2.5 | 0.81 |
| Summer dates | 4780 | −0.2 | −7 | 1.2 | 0.77 |
| Fall dates[∗] | 9643 | 1.2 | 24 | 4.6 | 0.65 |
| Winter dates | 3957 | 2.4 | 66 | 3.7 | 0.53 |
| Spring dates | 7576 | −0.9 | −26 | 2.0 | 0.69 |
| All dates[∗] | 25956 | 0.5 | 13 | 3.4 | 0.65 |

provides retrospective justification for increasing the CAMS-REG anthropogenic $NO_x$ emissions by a factor of 1.5 for all dates, as proposed in Sect. 2.1.2. While finer, day-specific adjustments based on column evaluations could be considered, they would likely introduce abrupt and potentially unrealistic temporal variations in emissions, e.g., in November 2021, when the mean model bias ranges from −5% to +125% across different days.

The RMSE exceeds $5 \times 10^{15}$ molec. cm$^{-2}$ on only three of the selected dates and remains at or below $2.5 \times 10^{15}$ molec. cm$^{-2}$ for 12 dates. The NMB lies within ±50% for 13 dates, and within ±25% for 9 dates, making them comparable to the results





obtained by Poraicu et al. (2023). Correlation coefficients range from 0.56 to 0.94, with 10 dates above 0.75 and a satisfactory overall value of 0.65 for the compilation of all selected dates.

The seasonal statistics in Table 7 show an underestimation of $NO_2$ columns during summer and spring, and an overestimation during winter and fall. The model underestimation in summer and spring is consistent with the underestimation of the observed surface concentrations during daytime (Sect. 3.1.2). These discrepancies may result from emission errors, inaccuracies in vertical mixing and/or oxidant levels, and possibly issues related to other model species. The surface measurements indicated a close agreement with the model during the first hours of daytime in fall and an overestimation in winter before the underestimation sets in (see Fig. 6). This does not appear in the comparison with SWING+ data in Table 7. One possible explanation is that the model lifts $NO_2$ species too far from the surface, at altitudes where SWING+ is more sensitive.

## 3.3 TROPOMI validation

### 3.3.1 Correcting WRF-Chem bias using SWING+

We first compare SWING+ measurements $\Omega_S$ with WRF-Chem outputs $\Omega_{W,S}$, accounting for SWING+ averaging kernels and acquisition times, but this time at TROPOMI spatial resolution. Specifically, we use a linear regression, denoted by $LR_1$, to predict the SWING+ column value from a given WRF-Chem column $\Omega_{W,S}$, as defined in Eq. (4):

$$LR_1(\Omega_{W,S}) = \alpha_0 + \alpha_1 \Omega_{W,S}, \tag{6}$$

where $\alpha_0$ and $\alpha_1$ are scalar values to be determined through separate linear regressions for each flight day. This is because the comparisons of WRF-Chem with SWING+ data show significant variations across different flight dates. Additionally, we exclude the flight of 22/11/2021 from the present analysis due to the lack of correlation between the model and the flight data (cf. Sect. 3.2).

We adopt the Theil-Sen estimator (Theil, 1950; Sen, 1968) for all selected dates (implemented via scipy.theilslopes along with a custom code to bootstrap the associated uncertainties). This method offers greater robustness to outliers and improved accuracy in error estimation compared to parametric methods such as ordinary and weighted least squares (Wilcox, 2010). A comparison of these three methods applied to our datasets is provided in Supplement 2. The results of the Theil-Sen regression for the selected flight dates are shown in Fig. 9. As expected from the results presented in Sect. 3.2, both intercepts and slopes vary significantly across the different flights, along with their associated uncertainties.

The WRF-Chem tropospheric columns used for comparison with TROPOMI, denoted $\Omega_{W,T}$, are constructed using TROPOMI averaging kernels and calculated at the satellite acquisition times, as defined in Eq. (5). However, these are likely biased, just like $\Omega_{W,S}$, so we define a bias-corrected version of the column, $\Omega_{W,T}^{bc}$, based on the model evaluation against SWING+ data, as derived in the previous step:

$$\Omega_{W,T}^{bc} = LR_1(\Omega_{W,T}) = \alpha_0 + \alpha_1 \Omega_{W,T}. \tag{7}$$

These bias-corrected columns can then be directly compared to the TROPOMI measurements, $\Omega_T$, since they are evaluated at the same time and account for TROPOMI vertical sensitivity through the term $\alpha_1 \Omega_{W,T}$. Note that the constant term, $\alpha_0$, was





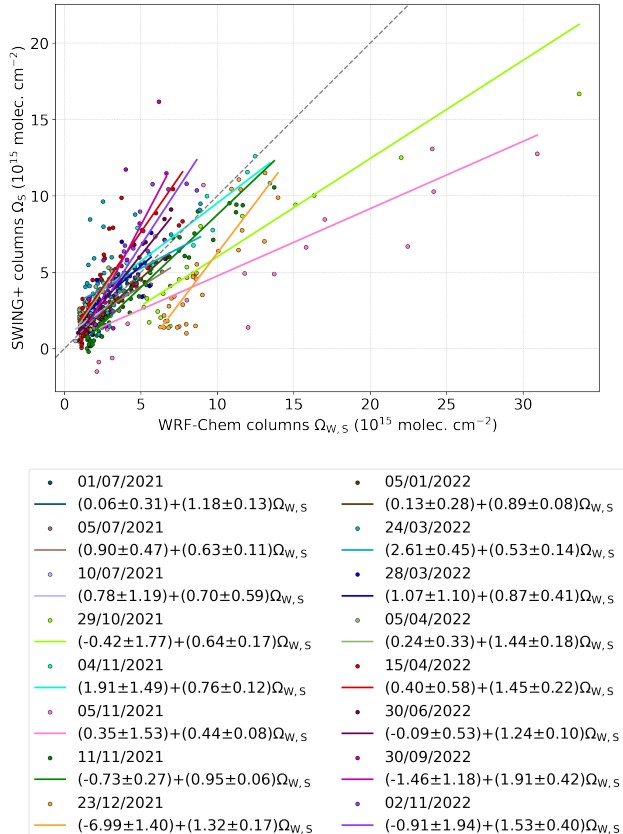

**Figure 9.** Scatter plot of SWING+ and WRF-Chem column values for our selection of 16 flight days. For each date, Theil-Sen estimators are used to determine the linear relationship $\mathrm{LR}_1$, along with associated uncertainties on the fitted coefficients.

evaluated while accounting for SWING+ vertical sensitivity. However, correcting this term is not feasible without additional
425 knowledge of the true atmospheric vertical profile. Nevertheless, its contribution to the overall expression is expected to be minor, as explained in Appendix A.

### 3.3.2 Evaluation of TROPOMI bias

By combining datasets from different flight days, either collectively or by season, we can assess the TROPOMI columns $\Omega_\mathrm{T}$ against the bias-corrected WRF-Chem columns $\Omega_\mathrm{W,T}^\mathrm{bc}$, using a linear regression denoted by $\mathrm{LR}_2$:

$$\mathrm{LR}_2(\Omega_\mathrm{W,T}^\mathrm{bc}) = \beta_0 + \beta_1 \Omega_\mathrm{W,T}^\mathrm{bc}, \tag{8}$$

where $\beta_0$ and $\beta_1$ are scalar parameters. Unlike in Sect. 3.3.1, this linear regression involves two datasets that both contain random errors. TROPOMI measurements are affected by instrument precision, with an average uncertainty of $\sigma_\mathrm{T} = 1.3 \times$



$10^{15}$ molec. cm$^{-2}$ across all selected dates. The average uncertainty of the bias-corrected dataset is limited by the precision of regression method used to produce it and is estimated at $\sigma_{\mathrm{LR}_1} = 0.5 \times 10^{15}$ molec. cm$^{-2}$.

Because most of the uncertainty is due to the TROPOMI columns,[1] the Theil-Sen estimator remains applicable in this context. We compare this approach to other parametric alternatives in Supplement 2. Among them, the orthogonal distance regression with weights (implemented via scipy.ODRPACK, Boggs et al., 1992) accounts explicitly for errors on both axes, together with possible heterogeneity (heteroscedasticity). As shown in Supplement 2, it produces similar regression results to the Theil-Sen method and performs slightly better in terms of mean absolute deviation of the fit. We interpret this result as
evidence that outliers do not significantly influence the orthogonal regression. Therefore, we choose this parametric method based on its better fit performance, while noting that both approaches yield consistent results and thus reinforce each other. The resulting scatter plot is shown in Fig. 10.

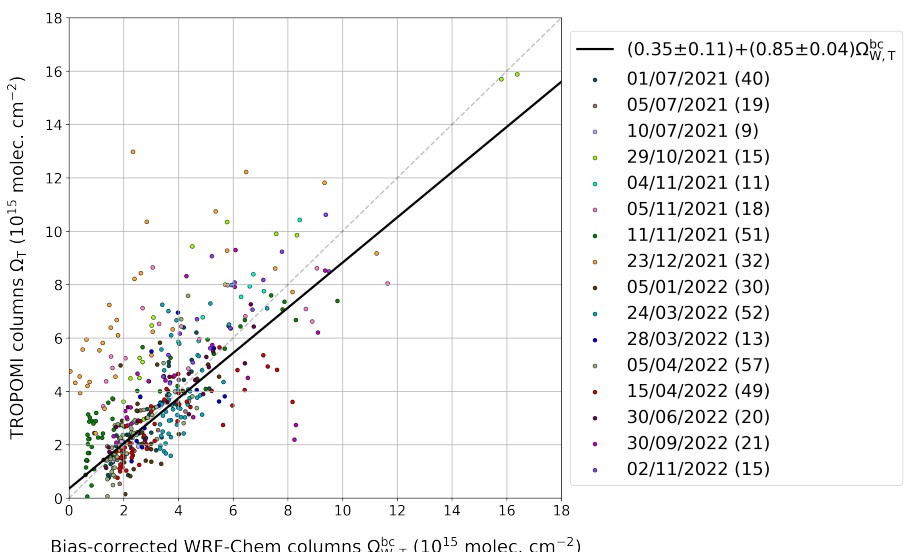

**Figure 10.** Scatter plot of 452 TROPOMI and bias-corrected WRF-Chem column values for all flight days (except 22/11/2021). Weighted orthogonal distance regression estimators are used to determine the linear relationship LR$_2$, along with associated uncertainties on the fitted coefficients. For each date, the number of columns is displayed in parentheses.

For different values of the bias-corrected column $\Omega_{\mathrm{T}}^{\mathrm{bc}}$ in the range $(1-15) \times 10^{15}$ molec. cm$^{-2}$, the regression line LR$_2$ allows to estimate the bias in TROPOMI measurements:

$$\Omega_{\mathrm{T}} - \Omega_{\mathrm{T}}^{\mathrm{bc}} = \mathrm{LR}_2(\Omega_{\mathrm{T}}^{\mathrm{bc}}) - \Omega_{\mathrm{T}}^{\mathrm{bc}} = \beta_0 + (\beta_1 - 1)\Omega_{\mathrm{T}}^{\mathrm{bc}} \pm \sigma_{\mathrm{b}} . \tag{9}$$

Before substituting numerical values into this expression, we first summarize the sources of uncertainty that contribute to the bias estimation, captured in $\sigma_{\mathrm{b}}$.

---

[1]The factor $\beta_1 \sigma_{\mathrm{LR}_1}/\sigma_{\mathrm{T}}$ governs the relative contribution of the uncertainties. Our assumption is supported by the small value of $\sigma_{\mathrm{LR}_1}/\sigma_{\mathrm{T}} = 0.38$ and a posteriori by the estimated regression line yielding $\beta_1 = 0.85$, as presented later in the text, such that $\beta_1 \sigma_{\mathrm{LR}_1}/\sigma_{\mathrm{T}} = 0.33$.



**Table 8.** TROPOMI mean biases, $MB = \Omega_T - \Omega_T^{bc}$ ($10^{15}$ molec. cm$^{-2}$), and relative biases, $RB = (\Omega_T - \Omega_T^{bc})/\Omega_T^{bc}$ (%), for various column values $\Omega_T^{bc}$ ($10^{15}$ molec. cm$^{-2}$) within the range of applicability of our results, roughly $(1-15) \times 10^{15}$ molec. cm$^{-2}$.

| $\Omega_T^{bc}$ | 1 | 2 | 4 | 6 | 8 | 10 | 12 | 15 |
|---|---|---|---|---|---|---|---|---|
| MB | $0.2 \pm 0.5$ | $0.0 \pm 0.6$ | $-0.3 \pm 0.7$ | $-0.6 \pm 0.9$ | $-0.9 \pm 1.2$ | $-1.2 \pm 1.4$ | $-1.5 \pm 1.7$ | $-1.9 \pm 2.0$ |
| RB | $20 \pm 52$ | $2 \pm 28$ | $-6 \pm 18$ | $-9 \pm 16$ | $-11 \pm 15$ | $-12 \pm 14$ | $-12 \pm 14$ | $-13 \pm 13$ |

The bias-corrected WRF-Chem columns $\Omega_{W,T}^{bc}$ implicitly account for the random error associated with the SWING+ columns, denoted $\sigma_{S,rand}$, through the regression $LR_1$ of Sect. 3.3.1. This random error is therefore reflected in the uncertainties of $LR_2$, as displayed in the legend of Fig. 10. However, the systematic component of the SWING+ measurement error, denoted $\sigma_{S,syst}$, was not included. We incorporate it now into the evaluation of TROPOMI bias, in addition to the random uncertainty already present from the evaluation of the regression $LR_2$, denoted as $\sigma_{LR_2}$:

$$\sigma_b = \sqrt{\sigma_{LR_2}^2 + \beta_1^2 \sigma_{S,syst.}^2} . \tag{10}$$

The uncertainty $\sigma_{LR_2}$ is determined from the uncertainties in the regression parameters $\beta_0$ and $\beta_1$, whereas $\sigma_{S,syst}$ arises from uncertainties associated with the reference slant column and the air mass factors used in the computation of the SWING+ vertical column density (Sect. 2.2.3). The error in the residual slant column is indeed purely systematic, and for simplicity, the AMF uncertainty is likewise assumed to be systematic, without a random component. Finally, we express the main result of this section as:

$$\Omega_T - \Omega_T^{bc} = 0.35 - 0.15\,\Omega_T^{bc} \pm \sqrt{(0.51)^2 - 0.01\,\Omega_T^{bc} + (0.13\,\Omega_T^{bc})^2} , \tag{11}$$

with $\Omega_T^{bc}$ in units of $10^{15}$ molec. cm$^{-2}$. Bias estimates for various column values $\Omega_T^{bc}$ are presented in Table 8. Details on how to obtain the numerical expression for the error from Eq. (10) are provided in Appendix B.

We can further invert the linear relation $LR_2$ to estimate a bias-corrected column $\Omega_T^{bc}$ for a given TROPOMI measurement $\Omega_T$, in $10^{15}$ molec. cm$^{-2}$:

$$\Omega_T^{bc} = -0.42 + 1.18\,\Omega_T \pm \sqrt{(0.61)^2 - 0.04\,\Omega_T + (0.19\,\Omega_T)^2} . \tag{12}$$

Similar to the previous expression, the uncertainty has been calculated to account for the systematic error in the SWING+ product, in addition to the uncertainty arising from the precision of the linear regression method.

We reproduce the linear regression for the selected dates grouped by season in Fig. 11. Our first remark is that the results for winter are of lower quality than in other seasons, due to the small size of the dataset, which covers only two dates (23/12/2021 and 05/01/2022) for a total of 62 columns. When focusing on the most reliable of the two dates, 05/01/2022, as identified in Table 7, we find that the resulting fit matches well the general relationship of Fig. 10. Therefore, we consider this date alone to provide a more robust basis for the winter analysis presented in the next paragraph. Note that excluding 23/12/2021 from the general analysis in Fig. 10 does not significantly affect the result. The resulting regression line becomes $(0.21 \pm 0.10) + (0.87 \pm 0.04)\Omega_{W,T}^{bc}$, which remains consistent with the original fit within the estimated uncertainties.



Remarkably, the summer scatter plot shows very little bias, with a value of $-0.1 \times 10^{15}$ molec.cm$^{-2}$, and no apparent dependence on the column value. Taking into account SWING+ systematic errors, at low column densities of $2 \times 10^{15}$ molec. cm$^{-2}$, we find relative biases of $-6 \pm 25\%$, $50 \pm 38\%$, $-15 \pm 44\%$, and $-14 \pm 24\%$ for summer, fall, winter, and spring, respectively. For high column values of $10^{16}$ molec. cm$^{-2}$ (even though this is slightly outside the range of applicability for summer, winter, and spring), we estimate the relative biases to be $-1 \pm 17\%$, $0 \pm 16\%$, $-15 \pm 29\%$, and $-18 \pm 14\%$, respectively.

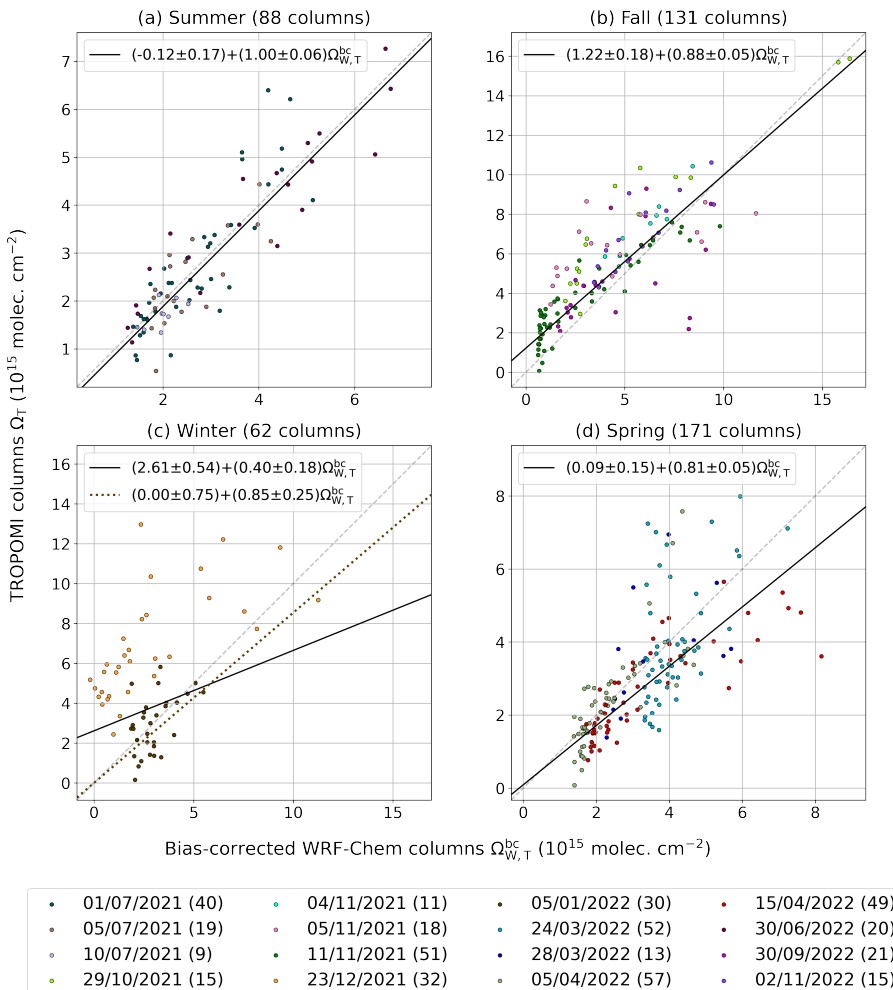

**Figure 11.** Seasonal scatter plots of TROPOMI versus bias-corrected WRF-Chem column values for the selected flight days: (a) summer, (b) fall, (c) winter, and (d) spring. Weighted orthogonal distance regression estimators are used to determine the seasonal linear relationships (solid lines), as well as the day-specific regression for 05/01/2022 during winter (dotted line), including the associated uncertainties on the fitted coefficients. For each time period, the number of columns is displayed in parentheses.





**Table 9.** Compilation of past studies evaluating TROPOMI tropospheric $NO_2$ against reference columns: $\Omega_P$ (Pandora) and $\Omega_{MD}$ (MAX-DOAS), in units of $10^{15}$ molec. cm$^{-2}$. The validation method used is either a direct comparison (a) or a comparison accounting for recalculated air mass factors (b). From the regressions, percentage relative biases (RB) at low ($4 \times 10^{15}$ molec. cm$^{-2}$) and high ($15 \times 10^{15}$ molec. cm$^{-2}$) column values are calculated.

| | Reference | TROPOMI product | Method | Regression line | RB, low col. | RB, high col. |
|---|---|---|---|---|---|---|
| **Pandora** | Griffin et al. (2019) | OFFL v1.1 | a | $-0.39 + 0.69\Omega_P$ | −41 | −34 |
| | | | b | $-0.26 + 0.74\Omega_P$ | −32 | −28 |
| | Zhao et al. (2020) | OFFL v1.1, 1.2$^{(rural)}$ | a | $1.10\Omega_P$ | 10 | 10 |
| | | | b | $1.15\Omega_P$ | 15 | 15 |
| | | OFFL v1.1, 1.2$^{(urban)}$ | a | $0.72\Omega_P$ | −28 | −28 |
| | | | b | $0.88\Omega_P$ | −12 | −12 |
| | Judd et al. (2020) | RPRO v1.2 | a | $-0.70 + 0.80\Omega_P$ | −38 | −25 |
| | | | b | $-0.20 + 0.82\Omega_P$ | −23 | −19 |
| **MAX-DOAS** | Dimitropoulou et al. (2020) | OFFL v1.2$^{(winter)}$ | a | $-0.27 + 0.81\Omega_{MD}$ | −26 | −21 |
| | | | b | $-5.09 + 1.67\Omega_{MD}$ | −60 | 33 |
| | | RPRO, OFFL v1.2$^{(spring)}$ | a | $1.86 + 0.47\Omega_{MD}$ | −6 | −41 |
| | | | b | $0.40 + 1.19\Omega_{MD}$ | 29 | 22 |
| | | RPRO, OFFL v1.2$^{(summer)}$ | a | $1.28 + 0.58\Omega_{MD}$ | −10 | −33 |
| | | | b | $-0.38 + 1.21\Omega_{MD}$ | 11 | 18 |
| | | RPRO, OFFL v1.2$^{(fall)}$ | a | $1.31 + 0.61\Omega_{MD}$ | −6 | −30 |
| | | | b | $0.71 + 0.97\Omega_{MD}$ | 15 | 2 |
| | Cai et al. (2022) | OFFL v1.2, 1.3 | a | $-0.85 + 0.84\Omega_{MD}$ | −37 | −22 |
| | van Geffen et al. (2022) | OFFL v1.2, 1.3 | a | $0.80 + 0.48\Omega_{MD}$ | −32 | −47 |
| | | DDS v2.1, 2.2 | a | $1.00 + 0.53\Omega_{MD}$ | −22 | −40 |
| | Yombo Phaka et al. (2023) | OFFL v2.1, 2.2, PAL v2.3 | a | $-0.21 + 0.67\Omega_{MD}$ | −38 | −34 |
| | | | b | $1.15 + 0.64\Omega_{MD}$ | −7 | −28 |

## 4 Review of TROPOMI tropospheric NO₂ validation

Tables 9 and 10 summarize literature results on the validation of TROPOMI tropospheric $NO_2$ products. The studies span from 2019 to 2025 and focus primarily on populated regions in North America, Europe, and China, while also including less-studied environments such as Kinshasa (Yombo Phaka et al., 2023). Table 9 compiles results from studies that employed Pandora column measurements from the Pandonia Global Network (Herman et al., 2009) and MAX-DOAS instruments (Hönninger et al., 2004), while Table 10 presents comparisons with airborne measurements, including our own results using SWING+. Overall,

the reported relative biases are predominantly negative across both low and high column abundances. Most biases fall within the ±50% acceptance range set by TROPOMI requirements (van Geffen et al., 2024). Note that, compared to the previous



**Table 10.** Same as for Table 9, except that airborne measurements are used for validation. The validation method used is either a direct comparison (a), a comparison accounting for recalculated air mass factors (b), or denotes the use of WRF-Chem combined with TROPOMI averaging kernels as detailed in Sect. 2 (c).

| | Reference | TROPOMI version | Method | Regression line | RB, low col. | RB, high col. |
|---|---|---|---|---|---|---|
| | Griffin et al. (2019) | OFFL v1.1 | a | $-0.26 + 0.89\Omega_A$ | $-18$ | $-13$ |
| | | | b | $-0.44 + 1.04\Omega_A$ | $-7$ | $1$ |
| | Judd et al. (2020) | RPRO v1.2 | a | $0.60 + 0.68\Omega_A$ | $-17$ | $-28$ |
| | | | b | $0.70 + 0.77\Omega_A$ | $-5$ | $-18$ |
| | Tack et al. (2021) | OFFL v1.3$^{(summer)}$ | a | $0.29 + 0.82\Omega_A$ | $-11$ | $-16$ |
| | | | b | $0.46 + 0.92\Omega_A$ | $3$ | $-5$ |
| Airborne | Lange et al. (2023) | OFFL v1.3 | a | $2.54 + 0.38\Omega_A$ | $2$ | $-45$ |
| | | | b | $2.36 + 0.41\Omega_A$ | $0$ | $-43$ |
| | | PAL v2.3 | a | $1.71 + 0.83\Omega_A$ | $26$ | $-6$ |
| | Poraicu et al. (2023) | OFFL v1.3$^{(summer)}$ | c | $0.64 + 0.82\Omega_A$ | $-2$ | $-14$ |
| | | PAL v2.3$^{(summer)}$ | c | $0.41 + 0.95\Omega_A$ | $5$ | $-2$ |
| | Johnson et al. (2023) | PAL v2.3 | a | $1.80 + 0.58\Omega_A$ | $3$ | $-30$ |
| | This work | RPRO v2.4$^{(winter)}$ | c | $0.00 + 0.85\Omega_A$ | $-15$ | $-15$ |
| | | RPRO v2.4$^{(spring)}$ | c | $0.09 + 0.81\Omega_A$ | $-17$ | $-18$ |
| | | RPRO v2.4$^{(summer)}$ | c | $-0.12 + 1.00\Omega_A$ | $-3$ | $-1$ |
| | | RPRO, OFFL v2.4$^{(fall)}$ | c | $1.22 + 0.88\Omega_A$ | $19$ | $-4$ |
| | | RPRO, OFFL v2.4 | c | $0.35 + 0.85\Omega_A$ | $-6$ | $-13$ |

section, we have reduced the column concentration range by raising the lower bound from $10^{15}$ to $4 \times 10^{15}$ molec. cm$^{-2}$. This adjustment reflects the fact that most of the referenced studies were conducted in polluted urban environments, typically more polluted than Bucharest and its surrounding rural regions, with NO$_2$ columns generally much higher than $10^{15}$ molec. cm$^{-2}$.

Results from Chan et al. (2020), Verhoelst et al. (2021), and Lambert et al. (2025), which do not fit our table format, are discussed separately below.

The reported studies span TROPOMI product versions from v1.1 to v2.8, with Lambert et al. (2025) using v2.4, the version adopted in this work. We summarize the changes following van Geffen et al. (2024), which guides our treatment of the different products. Versions v1.2 and v1.3 introduced only minor refinements with negligible impact on column values with respect to

v1.1; we group these together. A major update came with v1.4, which improved the cloud retrieval algorithm and led to higher NO$_2$ columns, especially in polluted regions. Further updates from v2.2 to v2.4 included improvements to the surface albedo, which enhanced radiative closure and reduced low biases in vegetated regions such as the Amazon basin. Versions v2.4 to v2.8 maintained a stable retrieval framework with only minor adjustments. Given the absence of v1.4 in our tables, and the overall consistency of later versions in urban settings, we group versions v2.1 through v2.8 together for comparison.



For v1.1 to v1.3, direct comparisons with Pandora, MAX-DOAS, and airborne measurements, indicate a median bias of $-17.5\%$ for low columns ($4 \times 10^{15}$ molec. cm$^{-2}$) and $-28\%$ for high columns ($15 \times 10^{15}$ molec. cm$^{-2}$). These results align with those of Verhoelst et al. (2021), who reported biases ranging from $-15\%$ to $-56\%$ in direct comparisons with MAX-DOAS across multiple sites worldwide.

Several studies recalculated the air mass factor (AMF) for versions v1.1 to v1.3 using alternative a priori profiles in place

of the a priori profiles of the TROPOMI data based on the TM5-MP model (Williams et al., 2017). For example, Griffin et al. (2019) and Zhao et al. (2020) used GEM-MACH profiles (Moran et al., 2010; Pendlebury et al., 2018); Judd et al. (2020) used NAMCMAQ (Stajner et al., 2011); and Tack et al. (2021) and Lange et al. (2023) used CAMS profiles (Colette et al., 2024). In addition, Chan et al. (2020) and Dimitropoulou et al. (2020) employed vertical profiles derived directly from MAX-DOAS observations. These adjustments generally lead to less negative, or more positive biases. For low columns, the median bias

across these studies is $-2.5\%$, and for high columns, $-2\%$. Chan et al. (2020) also noted that improving the AMF reduced the bias by up to 17%. In most of the reported studies, recalculating the AMF reduces the bias, with reductions of 5 to 20% observed in half of the cases.

The same aircraft campaign and TROPOMI product version (v1.3) were used by Tack et al. (2021) and Poraicu et al. (2023). Tack et al. (2021) reported results based on direct comparisons and using CAMS-based AMFs, while Poraicu et al. (2023)

aligned with our approach, employing the WRF-Chem model as an intercomparison platform and incorporating TROPOMI averaging kernels. The improvement relative to direct comparison is more pronounced when CAMS-based AMFs are used. Specifically, applying CAMS AMFs and the model-based method reduces the low-column bias from $-11\%$ to $3\%$ and $-2\%$, respectively. For high columns, the bias improves from $-16\%$ to $-5\%$ and $-14\%$.

We now turn to the evaluation of TROPOMI products v2.1 to v2.8. Median biases reported in Table 9 for direct comparisons

relative to MAX-DOAS are $-30\%$ for low columns and $-37\%$ for high columns. These values closely match those reported by Lambert et al. (2025): $-29\%$ for polluted stations (3 to $14 \times 10^{15}$ molec. cm$^{-2}$) and $-38\%$ for extremely polluted stations ($> 15 \times 10^{15}$ molec. cm$^{-2}$). Yombo Phaka et al. (2023) recalculated TROPOMI columns using vertical profiles derived from MAX-DOAS measurements and found a bias reduction of 31% and 6 % of low and high columns, respectively. Similarly, Lambert et al. (2025) noted that applying TROPOMI averaging kernels to MAX-DOAS vertical profiles leads to a bias reduction by up

to 20%.

Direct comparisons with aircraft campaigns indicate that biases for high columns have decreased with newer TROPOMI product versions. Using version v1.3, median biases are $-14\%$ for low columns and $-22\%$ for high columns. In contrast, for more recent versions (v2.3), we find similar low-column biases ($-14.5\%$), but improved performance for high columns, with a median bias of $-18\%$. This suggests that product upgrades have slightly improved performance in polluted conditions.

However, incorporating WRF-Chem and TROPOMI averaging kernels has a stronger impact, reducing the biases in version v2.3 to $5\%$ and $-2\%$ for low and high column values, respectively, as shown by Poraicu et al. (2023). Our summertime results using v2.4 are similar, with very small biases for low and high columns (Table 10). However, our low-column biases range from $-17\%$ to $19\%$ across seasons, and high-column biases may reach up to $-18\%$. Considering all seasons, overall biases are $-6\%$ for low columns and $-13\%$ for high columns in our work.



## 5 Conclusions

This study presents an evaluation of tropospheric $NO_2$ over Bucharest, combining high-resolution WRF-Chem simulations with multiple observational datasets. We assess WRF-Chem performance against in situ meteorological and surface concentration measurements, as well as airborne column observations from SWING+, while independently validating TROPOMI tropospheric $NO_2$ products using a model-based intercomparison framework. This joint analysis provides insight into both the modeling capabilities and satellite product validity over a complex and understudied urban environment.

Comparison against surface meteorological variables shows that WRF-Chem reproduces key features of regional meteorology. Across 17 two-day periods, surface pressure, temperature, relative humidity, and solar radiation are well represented, with mean biases within 1 mbar, 0.5°C, 6%, and 37 W m$^{-2}$, respectively. Temporal correlation coefficients are higher than 0.95 for pressure, temperature, and radiation, and higher than 0.85 for relative humidity. Wind speed exhibits a positive bias of 1.0 m s$^{-1}$, consistent with previous WRF-Chem studies (Kim et al., 2013; Feng et al., 2016; Poraicu et al., 2023), while wind direction shows a mean bias below 16°. The temporal correlation for the horizontal wind vector is generally weaker ($r = 0.64$). On 22/11/2021, a mismatch in wind direction appeared to negatively impact the modeled $NO_2$ column evaluation. Aside from this case, the model successfully captures the meteorological conditions required to support atmospheric chemistry assessments, using a common configuration and set of parameterizations.

Modeled surface concentrations of NO and $NO_2$ exhibit consistent daytime underestimations, concomitant with an overestimation of $O_3$. When restricting the comparison to non-traffic sites, the mean bias remains within $-8$ $\mu$g m$^{-3}$ for both NO and $NO_2$, accounting for potential interference from $NO_y$ reservoir species. Temporal correlations exceed 0.70 for NO and $NO_2$, and reach 0.81 for $O_3$, successfully capturing the diurnal and seasonal cycles of all three species. This agreement is improved for NO and $NO_2$ during colder months, and for $O_3$ during warmer periods.

WRF-Chem performs generally well against airborne SWING+ measurements of the tropospheric $NO_2$ column. Across 16 selected flight days, it exhibits a mean bias of $0.5 \times 10^{15}$ molec. cm$^{-2}$ (13%), with correlation coefficients exceeding 0.75 in 9 cases. Seasonal patterns emerge: summer and spring flights show model underestimation of $-7\%$ and $-26\%$, respectively, while fall and winter show positive biases of 24% and 66%. These results support an empirical upscaling of CAMS-REG v7.0 anthropogenic $NO_x$ emissions by a factor of 1.5 over Bucharest, which improves agreement with both surface and column measurements. The spring and summer underestimations of $NO_2$ columns are reminiscent of the surface underestimations observed during flight hours. However, a discrepancy arises in fall and winter, as the surface and SWING+ instruments exhibit opposite biases. Finally, we point to model improvements that could help reconcile surface and column levels, beyond correcting the emission inventory, and should be evaluated using more observational data. In particular, vertical mixing (especially in fall and winter) and processes affecting oxidant levels (e.g., volatile organic compounds and their photochemical oxidation) will require further attention.

TROPOMI tropospheric $NO_2$ columns v2.4.0 (RPRO+OFFL) are validated using bias-corrected model columns, with SWING+ serving as the reference and TROPOMI averaging kernels applied to the model profiles. The linear relationship expressing the original TROPOMI column, $\Omega_T$, in terms of its bias-corrected counterpart, $\Omega_T^{bc}$, is given by $\Omega_T = 0.35 + 0.85\,\Omega_T^{bc}$,





in units of $10^{15}$ molec. cm$^{-2}$. Relative biases vary with column magnitude, ranging from $20\%$ at $10^{15}$ molec. cm$^{-2}$ to $-13\%$

at $15 \times 10^{15}$ molec. cm$^{-2}$. A careful treatment of uncertainties from SWING+ observations and the regression method shows that relative bias errors are large at low column values (approximately $50\%$), but decrease to within $20\%$ for columns above $4 \times 10^{15}$ molec. cm$^{-2}$ and within $15\%$ for columns exceeding $8 \times 10^{15}$ molec. cm$^{-2}$. Seasonal analysis reveals greater variability in biases at low column values, ranging from $-17\%$ in spring to $19\%$ in fall. In contrast, higher column values exhibit more consistent negative biases, ranging from $-18\%$ in spring to $-1\%$ in summer. Overall, our results are in agreement with

findings from validation studies in the literature, particularly when considering the associated uncertainties and the methodology employed. Our literature review, focusing on studies over polluted areas, indicates that reported TROPOMI biases are predominantly negative. Recalculating air mass factors or applying TROPOMI averaging kernels often reduces these biases by approximately 5 to 20%, regardless of the version of the TROPOMI products used.

## Appendix A: Reference columns and vertical profiles

In Sect. 3.3.1, we introduced the vertical profile $n_{\mathrm{W}}$ modeled with WRF-Chem, where $z$ is the vertical coordinate. We can write a general equation to relate it to the true atmospheric profile, denoted by $n$:

$$n(z) = \delta n(z) + \alpha\, n_{\mathrm{W}}(z)\,, \tag{A1}$$

where $\alpha$ is an unknown scalar parameter, and $\delta n$ represents the deviation from linearity. Unlike $n$ and $n_{\mathrm{W}}$, the function $\delta n$ may take negative values. At this stage, Eq. (A1) remains too general to be directly informative.

Formally, integrating the profiles $n$ and $n_{\mathrm{W}}$ over the troposphere (Trop), using the airborne instrument averaging kernels $A_{\mathrm{S}}$, defines the bias-exempt and modeled tropospheric columns, $\Omega_{\mathrm{S}}$ and $\Omega_{\mathrm{W,S}}$, respectively:

$$\Omega_{\mathrm{S}} = \int_{\mathrm{Trop}} A_{\mathrm{S}}(z)n(z)\mathrm{d}z\,, \quad \Omega_{\mathrm{W,S}} = \int_{\mathrm{Trop}} A_{\mathrm{S}}(z)n_{\mathrm{W}}(z)\mathrm{d}z\,. \tag{A2}$$

When a linear regression is performed on the datasets $\Omega_{\mathrm{S}}$ and $\Omega_{\mathrm{W,S}}$, we estimate the parameters $\alpha_0$ and $\alpha_1$ that define the regression line for the estimated values, $\mathrm{LR}_1$:

$$\mathrm{LR}_1(\Omega_{\mathrm{W,S}}) = \alpha_0 + \alpha_1 \Omega_{\mathrm{W,S}}\,. \tag{A3}$$

These parameters can now be used to constrain Eq. (A1) through the following relations:

$$\int_{\mathrm{Trop}} A_{\mathrm{S}}(z)\delta n(z)\mathrm{d}z = \alpha_0\,, \quad \alpha = \alpha_1\,. \tag{A4}$$

Together with our detailed knowledge of the modeled profile $n_{\mathrm{W}}$, this allows us to construct reference, or bias-corrected, columns for comparison with another instrument for which a bias must be estimated.

For the satellite instrument, these new modeled columns are denoted $\Omega_{\mathrm{W,T}}^{\mathrm{bc}}$ in the main text and are defined using the satellite averaging kernels $A_{\mathrm{T}}$:

$$\Omega_{\mathrm{W,T}}^{\mathrm{bc}} = \int_{\mathrm{Trop}} A_{\mathrm{T}}(z)n(z)\mathrm{d}z = \int_{\mathrm{Trop}} A_{\mathrm{T}}(z)\delta n(z)\mathrm{d}z + \alpha_1 \int_{\mathrm{Trop}} A_{\mathrm{T}}(z)n_{\mathrm{W}}(z)\mathrm{d}z\,. \tag{A5}$$



The first term in the expression above can be expanded around $\alpha_0$, while the second corresponds to the definition of $\Omega_{\mathrm{W,T}}$, as introduced in the main text:

$$\Omega_{\mathrm{W,T}}^{\mathrm{bc}} = \alpha_0 + \int_{\mathrm{Trop}} [A_{\mathrm{T}}(z) - A_{\mathrm{S}}(z)]\,\delta n(z)\mathrm{d}z + \alpha_1 \Omega_{\mathrm{W,T}} = \mathrm{LR}_1(\Omega_{\mathrm{W,T}}) + \int_{\mathrm{Trop}} [A_{\mathrm{T}}(z) - A_{\mathrm{S}}(z)]\,\delta n(z)\mathrm{d}z \,. \tag{A6}$$

Unfortunately, the last integral cannot be evaluated without more precise knowledge of $n$, and thus $\delta n$. In general, if the model performs well, $\delta n$ remains small in absolute value, and the extra integral can be neglected. In this specific case, we are further helped by the structure of the integrand: the kernel difference $A_{\mathrm{T}} - A_{\mathrm{S}}$ places greater weight on altitudes above the aircraft, where the true and modeled $NO_2$ concentrations, and therefore $\delta n$, are relatively low compared to those in the boundary layer
over polluted urban areas. As a result, the contribution of the extra integral to the overall expression is further suppressed. We therefore expect this term to be minor, and make the following approximation in the main text:

$$\Omega_{\mathrm{W,T}}^{\mathrm{bc}} = \mathrm{LR}_1(\Omega_{\mathrm{W,T}}) \,. \tag{A7}$$

As a side remark, if $\delta n$ is small in absolute value from the ground to the troposphere, then $\alpha_0$ should also be small. However, this assumption is stronger than what is required in the main text.

**Appendix B: Errors in TROPOMI bias estimation**


In Sect. 3.3.1, we estimate the bias of TROPOMI and its associated uncertainty, denoted by $\sigma_b$. We explain that this uncertainty is the quadrature sum of the random component from the linear regression, $\sigma_{\mathrm{LR}_2}$, and the propagated systematic error from the SWING+ measurements, $\sigma_{\mathrm{S,syst}}$ multiplied by the slope $\beta_1$. For clarity, we repeat its expression here:

$$\sigma_{\mathrm{b}} = \sqrt{\sigma_{\mathrm{LR}_2}^2 + \beta_1^2 \sigma_{\mathrm{S,syst}}^2} \,. \tag{B1}$$

The linear regression $\mathrm{LR}_2$ presented in Fig. 10 led to the estimation of the intercept and slope parameters, $\beta_0 = 0.35 \times 10^{15}$ molec. cm$^{-2}$ and $\beta_1 = 0.85$, with respective uncertainties $\sigma_{\beta_0} = 0.11 \times 10^{15}$ molec. cm$^{-2}$ and $\sigma_{\beta_1} = 0.04$. Additionally, the covariance between these two estimated parameters must be taken into account: $\sigma_{\beta_0 \beta_1} = -0.004 \times 10^{15}$ molec. cm$^{-2}$. The regression line was used to predict the TROPOMI column based on a given bias-corrected column $\Omega_{\mathrm{T}}^{\mathrm{bc}}$, which we denote in this section as $\Omega$ for clarity (in $10^{15}$ molec. cm$^{-2}$). The uncertainty of the predicted value is given by:

$$\sigma_{\mathrm{LR}_2}(\Omega) = \sqrt{\sigma_{\beta_0}^2 + 2\sigma_{\beta_0 \beta_1}\Omega + \sigma_{\beta_1}^2 \Omega^2} = \sqrt{(0.11)^2 - 0.01\,\Omega + (0.04\,\Omega)^2} \,. \tag{B2}$$

For predictor columns $\Omega$ equal to $10^{15}$ and $10^{16}$ molec. cm$^{-2}$, the resulting errors are $0.03 \times 10^{15}$ and $0.42 \times 10^{15}$ molec. cm$^{-2}$, respectively.

As explained in Sect. 3.3.1, we assume that the systematic error of SWING+ arises from the total errors on the reference slant column densities and air mass factors, propagated to the vertical column density $\Omega$. These components were presented in
Sect. 2.2.3. The first, denoted here as $\sigma_{\mathrm{S,ref}}$, averages to $0.58 \times 10^{15}$ molec. cm$^{-2}$ when considering all dates included in the



TROPOMI validation analysis, weighted by the number of columns per date. The second component, $\sigma_{S,AMF}$, is a relative error of 15.2% on the column $\Omega$, consistently applied across all dates.

$$\sigma_{S,syst}(\Omega) = \sqrt{\sigma_{S,ref}^2 + \sigma_{S,AMF}(\Omega)^2} = \sqrt{(0.58)^2 + (0.15\,\Omega)^2} \tag{B3}$$

This leads to systematic errors of $0.60 \times 10^{15}$ and $1.61 \times 10^{15}$ molec. cm$^{-2}$ for predictor columns $\Omega$ equal to $10^{15}$ and $10^{16}$ molec. cm$^{-2}$, respectively. Multiplying by $\beta_1$, we find the corresponding errors propagated to the predicted values: $0.51 \times 10^{15}$ and $1.37 \times 10^{15}$ molec. cm$^{-2}$, respectively.

Combining the previous expressions, we obtain the following equation, as presented in the main text:

$$\sigma_b(\Omega) = \sqrt{(0.51)^2 - 0.01\,\Omega + (0.13\,\Omega)^2}. \tag{B4}$$

The errors on the predicted values are $0.52 \times 10^{15}$ and $1.44 \times 10^{15}$ molec. cm$^{-2}$ for $\Omega$ equal to $10^{15}$ and $10^{16}$ molec. cm$^{-2}$, respectively. Note that most of the error originates from the propagated systematic uncertainty associated with the SWING+ measurements.

*Code and data availability.*  The WRF-Chem model and WPS codes are distributed by NCAR (https://www.mmm.ucar.edu/models/wrf, last access: 22 July 2025;  Skamarock et al., 2019). WRF-Chem processing tools are provided separately (https://www2.acom.ucar.edu/wrf-chem/wrf-chem-tools-community, last access: 22 July 2025). Python scripts used for regridding, column calculation, and statistical analysis are available upon request. Static geographical data used in WRF-Chem are provided by NCAR (https://www2.mmm.ucar.edu/wrf/users/download/get_sources_wps_geog.html, last access: 22 July 2025). ERA5 reanalysis data are distributed via the Climate Data Store from ECMWF (https://cds.climate.copernicus.eu/datasets, last access: 22 July 2025;  Hersbach et al., 2023a, b). The CAMS-REG anthropogenic emission inventory is available through the ECCAD catalogue (https://eccad.aeris-data.fr/, last access: 22 July 2025;  Kuenen et al., 2022). Meteorological measurements from the MARS station are accessible via the PANGAEA portal (https://dataportals.pangaea.de/bsrn/stations, last access: 22 July 2025;  Carstea et al., 2025). ANM measurements are available upon request through the MeteoRomania website (https://www.meteoromania.ro/, last access: 22 July 2025). RNMCA in situ measurements can be downloaded from the CalitateAer website (https://calitateaer.ro/, last access: 22 July 2025). SWING+ measurements are avaible upon request. TROPOMI NO$_2$ column data are available from the Copernicus Data Space (https://dataspace.copernicus.eu/, last access: 22 July 2025).

*Author contributions.*  AP conducted the simulations, prepared the necessary data, performed the comparisons, and wrote the draft of the paper. JFM and TS conceptualized the project, supervised the work, and aided in the interpretation of results. CP helped with computational requirements and advised on the simulations. AM and FT provided SWING+ measurements and guidance about their usage. All coauthors read and commented the manuscript.

*Competing interests.*  The contact author has declared that none of the authors has any competing interests.



*Acknowledgements.* We thank Raluca Smău for providing meteorological measurements from ANM stations.

*Financial support.* This work was supported by the Belgian Science Policy Office (BELSPO) through the European Space Agency-funded PRODEX TROVA-3 project (2024-2026).



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
