# Peer review of "Validation of TROPOMI and WRF-Chem NO2 across seasons using SWING+ and surface observations over Bucharest"

_EGUsphere, 2025_

## Author Comment (AC1)

**REPLY TO REFEREE #1**

We thank the referee for the thorough review of our manuscript, as well as for the careful evaluation, constructive suggestions, and positive feedback. Referee comments are reproduced in black, and our replies are given in blue.

Review of the study „Validation of TROPOMI and WRF-Chem NO2 across seasons using SWING+ and surface observations over Bucharest" by Antoine Pasternak et al.

The study focuses on validating a high-resolution chemistry-transport model and satellite retrievals of tropospheric nitrogen dioxide ($NO_2$) over the urban region of Bucharest. The authors run the WRF-Chem model with 1 km resolution over the Bucharest region for 17 two-day time series in 2021 and 2022 across the different seasons, and compare its output with ground-based in situ meteorological and chemical observations as well as airborne column measurements from the SWING+ instrument (17 flights between 2021–2022). They evaluate the satellite-based TROPOMI tropospheric $NO_2$ column product (v2.4.0) using the airborne SWING+ data as reference, and WRF-Chem as an intercomparison. The main findings are the different biases for different concentration ranges and the seasonality of the results. Results are compared to existing TROPOMI validation studies. They show that NOx emissions from the CAMS-REF inventory need to be scaled.

**General comments:**

The paper presents valuable and comprehensive validation work for both TROPOMI and WRF-Chem $NO_2$ data over the Bucharest region. It provides detailed comparisons combining satellite, model, airborne, and surface observations over an urban area with complex emission patterns. The analysis across different seasons and atmospheric conditions adds further strength.

I think it needs discussion, if not only a scaling of the CAMS-REG, but a seasonal dependent scaling is necessary. Can you also comment on how region-dependent your scaling factor is? I think the seasonal dependency in the validation results should, in general, be more highlighted in the abstract and conclusion.

We thank the referee for this invitation to further discussion. We acknowledge that a constant scaling factor is unlikely to capture the full complexity of emission variability. However, deriving credible seasonal scaling factors is beyond the scope of our analysis. This would require a robust mass-balance approach across the available flight dates to isolate seasonal emission effects, which is nontrivial given that some seasons are less represented and the influence of various factors besides emissions on $NO_2$ levels.

Moreover, our 1.5 factor is sufficient to close the gap with SWING+ magnitudes on average (see Table 6), which is adequate for our purpose of validating TROPOMI, since

the remaining day-to-day discrepancies are handled through the bias-correction procedure described in Section 3.3.1.

We note that the preprint by Hohenberger et al. (2025) indicates that CAMS-REG underestimates road-transport urban $NO_x$ emissions by about 35 % in European cities, based on a comparison with independent urban inventories. This is broadly consistent with our results, given the expected variability in uncertainties across cities, and helps explain why our factor applies to the Bucharest region.

Following the referee's advice, both this last comment and the seasonal dependency of TROPOMI bias have been highlighted in the conclusion. We also modified the abstract to add details on the seasonal analysis. These modifications are presented in response to the specific comments referring to lines L558 and L577/578 below.

*I think section 4 would benefit from being less of a review and including more explicit comparisons of this study's results with previous validation studies.*

Following the referee's advice, we have added details on our seasonal results in the context of the existing literature, where comparisons could be made. Please see our response to the specific comment referring to lines L577/578.

*Overall, this is a well-structured, technically sound paper, and its results are relevant to the air quality satellite and modelling community. I recommend publication after minor adjustments.*

**Specific comments:**

*L23: Missing reference.*

The reference of Seinfeld and Pandis (2016) has been added.

*L53: Add a reference for the lifetime information.*

The reference of Laughner et al. (2019) has been added.

*L55: Add a reference for the expected seasonality.*

The reference of Boersma et al. (2009) has been added.

*L101: What is this highest resolution available?*

The resolution varies between datasets; all are referenced on the website cited in the manuscript.

*L103: Which ERA5 variables have been used?*

We added a comprehensive list in Supplement 1.

*L110: Instead of "each chemical species" maybe better "several chemical species". Which have you used?*

The first paragraph of Sect. 2.1.2 has been updated to list the various chemical species included in the CAMS-REG inventory.

*L113: How do the monthly factors vary by season for the main sectors contributing to the emissions over Bucharest? Do you know how these factors are determined?*

The figure of CAMS-REG $NO_x$ sectoral emissions over Bucharest was replaced with a seasonal emission plot that retains sectoral information in Sect. 2.1.2. The text now highlights that the temporal factors from TNO rely primarily on energy consumption statistics and traffic counts (Denier van der Gon et al., 2011).

*Table 2: I would suggest putting this table in the attachment. How is the NO/NO2 ratio determined for the mapping of CAMS-REG NOx into MOZART-4 NO and NO2 and how is this influencing your results?*

We moved the Table to Supplement 2, next to the following comments.

"$NO_x$ species are distributed as 90% NO and 10% $NO_2$ following the MOZCART users' guide (https://www.acom.ucar.edu/wrf-chem/MOZCART_UsersGuide.pdf, last access: 14 January 2026). Different values were reported in the literature, with $NO_2/NO_x$ ratios ranging from 5.3% to 39% (Kuhn et al. (2024)). However, we expect this choice to have a negligible effect on the results, since NO has a very short lifetime during daytime (a few minutes; Seinfeld and Pandis (2016)) and photochemical equilibrium is rapidly established."

*L160: "The first data point is recorded at 01:00 LT." Why is this different from the meteorology comparison?*

We decided to keep 24 RNMCA data points per day, starting with the one recorded at 01:00 LT and effectively covering the first hour of each day to ensure daily representativeness. In comparison, MARS provides meteorological data at one-minute intervals, for which we retained 1440 measurements per day (60×24) to maintain daily representativeness.

*Equation 1: How were the composition and ratios determined?*

We added a note to emphasize that these ratios are prescribed by Lamsal et al (2008). They have also been used in previous modeling studies (e.g., Poraicu et al. (2023), Kuhn et al. (2024)).

*L216: Just to clarify, with these errors, you mean the AMF, SCDre, and DSCDs errors?*

We mention the contribution of AMF, SCD$_{ref}$, and DSCDs errors to the VCD error. This sentence has been revised to clarify our point:

"The combined contributions of the AMF, SCD$_{ref}$, and DSCD yield a total VCD uncertainty of 0.9-1.9 × $10^{15}$ molec. cm$^{-2}$."

*L226: You are integrating over the troposphere, but SWING+ only measured below 3km, is this correct?*

The light-path from the Sun to an airborne nadir instrument, such as SWING+, traverses the upper troposphere. As such, these measurements are sensitive to the entire troposphere, even though their sensitivity is highest below the aircraft altitude. Averaging kernels are defined over the full altitude range, in order to take these variations into account (Merlaud et al. (2018); Tack et al. (2021)).

*L318-320: Do you mean, it is expected to see a morning, a late afternoon and an evening peak? Because you see two peaks (morning and evening), which I thought are both related to the rush hour. But if I understand your discussion correctly, you expect an late afternoon rush hour peak? What is causing the evening peak?*

As explained by Poraicu et al. (2023), there is no late-afternoon peak due to the counterbalancing effects of chemistry and planetary boundary layer mixing. The evening peak occurs because emissions remain high (though much lower than in the late afternoon), while chemical loss and mixing are much weaker.

*L331: This might be because the second day is always the flight day, so usually a clear-sky day.*

This is correct. However, the original sentence was misleading, as we intended to emphasize that the second winter day shows better agreement compared to other seasons, rather than relative to the first day. We have revised the text accordingly: "The second day of the winter runs shows the best daytime agreement, relative to other seasons. However, because the winter analysis is based on only two time series, it is difficult to draw definitive conclusions regarding which season is best reproduced by the model."

*L373-376: Move "It also provides statistics per season and for the entire dataset. For two dates, reported in the table, we truncate data associated with the beginning of the flight for reasons explained in Sect. 3.2.2." to L373 after "for each separate flight.". Which is the other truncated day? Selecting 13:24 LT instead of what time?*

The paragraph has been revised following the referee' suggestion and adding some information on the truncated flight and start time.

*L385: Be careful in the discussion, and remind the reader that some seasonal biases might compensate each other.*

We adapted the paragraph accordingly:

"The comparable numbers of days with either positive (7) or negative (10) biases in Table 6 suggest a balanced model behavior on average. The small overall bias across all selected dates (MB of $0.5 \times 10^{15}$ molec. cm$^{-2}$ and RB of 13%), along with the

underestimation in surface $NO_2$* found in Sect. 2.2.2 (MB of -8 µg m$^{-3}$ and RB of -33%), provides a retrospective justification for increasing the CAMS-REG anthropogenic $NO_x$ emissions by a factor of 1.5, as proposed in Sect. 2.1.2.

The small overall model bias against SWING+ reflects compensating seasonal biases of opposite sign, indicating that a temporally varying scaling factor for $NO_x$ emissions may be more realistic. However, while finer, day-specific adjustments based on the column evaluations in Table 6 could be considered, they would likely introduce abrupt and potentially unrealistic temporal variations in emissions, e.g., in November 2021, when the mean model bias ranges from -5% to +125% across different days. This variability may reflect the fact that, in addition to emission uncertainties, the model daily performance (e.g., chemistry and transport) on a limited set of days can strongly influence seasonal statistics, particularly in winter and fall, whereas spring and summer appear more consistent."

*Table 7: Do you have an idea why the agreement is so different between the days?*

It is difficult to pinpoint the sources of model errors, as they can be numerous, including performance under specific meteorological conditions, boundary and initial conditions, or biases in emissions. The dramatic case of 22/11/2021 exemplifies a situation in which a wind direction change was poorly reproduced by the model, as discussed in the second paragraph of Section 3.2.3.

*Line 471: Do you have any ideas why this day you have excluded is not working well?*

We added a note in Sect. 3.3.2:

"The flight day of 23/12/2021 shows less convincing results (Table 6) and is characterized by consistently high modeled background values (see Supplement 4), which may be due to inaccurate initial or boundary conditions for $NO_x$ species, oxidant concentrations, and/or heterogeneous chemistry on aerosols."

This last possibility is hinted at by a significant difference in aerosol optical depth (AOD) values measured at the INOE station in Măgurele on 23/12/2021 compared with 05/01/2022, as shown in the plots below (top: 23/12/2021, bottom: 05/01/2022).

[Figure]

[Figure]

*Line 481: ...span from 2019 to 2025, cover several TROPOMI product versions and focus...*

Done.

*L487: and the upper bound from 10^16 to 1.5x10^16.*

The upper bound was already placed at $1.5 \times 10^{16}$ molec. cm$^{-2}$ in Sect. 3.3.2. We adapted the sentence to refer to the upper bound, for clarity.

*L558: See general comments. Please comment if seasonal scaling would be better and how region-dependent your factor might be.*

We have adapted the line referring to our scaling of CAMS-REG into a paragraph in the conclusion, following the referee's advice.

"The underestimation of WRF-Chem NO and $NO_2$ daytime surface levels, along with the small positive bias for $NO_2$ modeled column magnitudes across different flight dates, supports an empirical upscaling of CAMS-REG v7.0 anthropogenic $NO_x$ emissions over Bucharest. It is also consistent with the documented low bias in CAMS-REG road-traffic $NO_x$ emissions in European cities with respect to independent urban inventories, estimated at approximately -35% (Hohenberger et al., 2025). The factor of 1.5 was sufficient for our purpose of validating TROPOMI. However, for a more in-depth assessment of the CAMS-REG inventory, different temporal profiles could be tested (e.g., Guevara et al. (2021)), and the overall magnitude could be adjusted seasonally using mass-balance inversion techniques (e.g., (Cooper et al., 2017; Poraicu et al., 2023))."

*L577/578 Connect this to what you have found. How much seasonality was studied in these evaluations of the TROPOMI product? Might this be something that should be investigated further?*

We added some details in Sect. 4:

"Finally, we assess the seasonal dependence of the TROPOMI bias. In our study, low-column biases range from -17% to 19% across seasons, while high-column biases range from -1% to -18%. Our summer results (-3% and -1% for low and high columns, respectively) agree well with the aircraft-based analysis for the PAL v2.3 product of Poraicu et al. (2023), with differences of 8% or less. Our fall results (19% and -4%) are consistent with those of Dimitropoulou et al. (2020) using recalculated AMF, with differences within 6%. They are also in line with Lange et al. (2023), showing positive biases for low columns. However, for high columns in fall, both our study and Lange et al. (2023) report negative biases, a feature captured by Dimitropoulou et al. (2020) only when using the original AMF. In contrast, winter and spring results show weaker consistency with Dimitropoulou et al. (2020). However, differences in methodology (notably the use of dual-scan MAX-DOAS observations) and in the TROPOMI product version limit direct comparability. This underscores the need for further validation studies, particularly in winter and spring, where comparable aircraft campaigns are lacking."

This is linked to an addition in the conclusion:

"Good agreement is found with seasonal studies comparing TROPOMI with aircraft (summer) and MAX-DOAS (fall) measurements, with differences relative to our results below 10%. The scarcity of seasonal studies and the differences in methodology, however, limit the comparability and highlight the need for more dedicated validation campaigns, particularly in winter and spring."

We also added a line to the abstract addressing the seasonality of our results:

"Seasonal diagnostics indicate variability in the bias for low columns, showing a marked positive bias in fall and negative biases in other seasons, whereas the negative bias at higher columns remains stable."

***Technical corrections:***

*L6/7: Split the sentence into two: ...are underestimated. Satisfactory agreement with observations is achieved ...*

Done.

*L29: like the Global Ozone ...*

Done.

*L120: A preliminary evaluation ...*

Done.

*Fig.3: stationary instead of staionnary for sector C*

This figure has been adapted into a table in Supplement 2.

*L142: ...for the first 15 of the 17 SWING+ dates...*

Done.

*L143/144: Move the sentence "When available, ..." to the end of this subsection.*

This sentence is specific to the MARS measurements and so we left it untouched.

*Table 3: I would replace the 0 with "-", since the meaning is not that there are 0 overpasses but just no O3 measurements at these stations*

Done.

*L156: including NO and NO2, and for some of them also O3.*

Done.

*L201: Typical and typically so close to each other doesn't read very well.*

Done.

*L204: remove with*

Done.

*L237: DOAS was already introduced.*

Done.

*L245: and the offline product instead of and offline products*

Done.

*L252: To avoid confusions: Does "for its" refer to SWING+ or WRF-Chem?*

We replaced "its" with "WRF-Chem".

*L326: Add …modeled outputs, separated by seasons, defined as summer (June, July, August), …*

Done.

*L367: Thereafter, the modeled columns correlate…*

Done.

*L433: precision of the regression method*

Done.

*L476&478: For better readability you can try to add the seasons directly after each bias: -6+/-25% (summer), …*

Done.

*L504/505: Hard to read, check rewriting.*

We replaced "Several studies recalculated the air mass factor (AMF) for versions v1.1 to v1.3 using alternative a priori profiles in place of the a priori profiles of the TROPOMI data based on the TM5-MP model." with "Several studies recalculated the air mass factor (AMF) for versions v1.1 to v1.4 using alternative a priori profiles in place of those from the TM5-MP model…"

*L537: We assess the WRF-Chem performance…*

Done.

**References**

- Boersma, K. F., Jacob, D. J., Trainic, M., Rudich, Y., De Smedt, I., Dirksen, R., and Eskes, H. J.: Validation of urban NO2 concentrations and their diurnal and seasonal variations observed from the SCIAMACHY and OMI sensors using in situ surface measurements in Israeli cities, Atmospheric Chemistry and Physics, 9, 3867–3879, https://doi.org/10.5194/acp-9-3867-2009, 2009.

- Cooper, M., Martin, R. V., Padmanabhan, A., and Henze, D. K.: Comparing mass balance and adjoint methods for inverse modelling of nitrogen dioxide columns for global nitrogen oxide emissions, Journal of Geophysical Research:

Atmospheres, 122, 4718–4734, https://doi.org/https://doi.org/10.1002/2016JD025985, 2017.

- Denier van der Gon, H., Hendriks, C., Kuenen, J., Segers, A., and Visschedijk, A.: Description of current temporal emission patterns and sensitivity of predicted AQ for temporal emission patterns, Tech. rep., TNO, Princetonlaan 6, 3584 CB Utrecht, The Netherlands, https://atmosphere.copernicus.eu/sites/default/files/2019-07/MACC_TNO_del_1_3_v2.pdf, 2011.

- Dimitropoulou, E., Hendrick, F., Pinardi, G., Friedrich, M. M., Merlaud, A., Tack, F., De Longueville, H., Fayt, C., Hermans, C., Laffineur, Q., Fierens, F., and Van Roozendael, M.: Validation of TROPOMI tropospheric NO2 columns using dual-scan multi-axis differential optical absorption spectroscopy (MAX-DOAS) measurements in Uccle, Brussels, Atmospheric Measurement Techniques, 13, 5165–5191, https://doi.org/10.5194/amt-13-5165-2020, 2020.

- Guevara, M., Jorba, O., Tena, C., Denier van der Gon, H., Kuenen, J., Elguindi, N., Darras, S., Granier, C., and Perez Garcia-Pando, C.: Copernicus Atmosphere Monitoring Service TEMPOral profiles (CAMS-TEMPO): global and European emission temporal profile maps for atmospheric chemistry modelling, Earth System Science Data, 13, 367–404, https://doi.org/10.5194/essd-13-367-2021, 2021.

- Hohenberger, T. L., Malki, M. E., Visschedijk, A., Guevara, M., Ramacher, P., Marongiu, A., Lanzani, G. G., Fossati, G., Kousa, A., Athanasopoulou, E., Kakouri, A., and Kuenen, J.: Link-based European road transport emissions for CAMS-REG v8.1 and a comparison to city inventories, Earth System Science Data Discussions, 2025, 1–28, https://doi.org/10.5194/essd-2025-428, 2025.

- Kuhn, L., Beirle, S., Kumar, V., Osipov, S., Pozzer, A., Bosch, T., Kumar, R., and Wagner, T.: On the influence of vertical mixing, boundary layer schemes, and temporal emission profiles on tropospheric NO2 in WRF-Chem – comparisons to in situ, satellite, and MAX-DOAS observations, Atmospheric Chemistry and Physics, 24, 185–217, https://doi.org/10.5194/acp-24-185-2024, 2024.

- Lamsal, L. N., Martin, R. V., van Donkelaar, A., Steinbacher, M., Celarier, E. A., Bucsela, E., Dunlea, E. J., and Pinto, J. P.: Ground-level nitrogen dioxide concentrations inferred from the satellite-borne Ozone Monitoring Instrument, Journal of Geophysical Research: Atmospheres, 113, https://doi.org/10.1029/2007JD009235, 2008.

- Lange, K., Richter, A., Schonhardt, A., Meier, A. C., Bosch, T., Seyler, A., Krause, K., Behrens, L. K., Wittrock, F., Merlaud, A., Tack, F., Fayt, C., Friedrich, M. M.,

Dimitropoulou, E., Van Roozendael, M., Kumar, V., Donner, S., Dorner, S., Lauster, B., Razi, M., Borger, C., Uhlmannsiek, K., Wagner, T., Ruhtz, T., Eskes, H., Bohn, B., Santana Diaz, D., Abuhassan, N., Schuttemeyer, D., and Burrows, J. P.: Validation of Sentinel-5P TROPOMI tropospheric NO2 products by comparison with NO2 measurements from airborne imaging DOAS, ground-based stationary DOAS, and mobile car DOAS measurements during the S5P-VAL-DE-Ruhr campaign, Atmospheric Measurement Techniques, 16, 1357–1389, https://doi.org/10.5194/amt-16-1357-2023, 2023.

- Laughner, J. L. and Cohen, R. C.: Direct observation of changing NOx lifetime in North American cities, Science, 366, 723–727, https://doi.org/10.1126/science.aax6832, 2019.

- Merlaud, A., Tack, F., Constantin, D., Georgescu, L., Maes, J., Fayt, C., Mingireanu, F., Schuettemeyer, D., Meier, A. C., Schonardt, A., Ruhtz, T., Bellegante, L., Nicolae, D., Den Hoed, M., Allaart, M., and Van Roozendael, M.: The Small Whiskbroom Imager for atmospheric compositioN monitorinG (SWING) and its operations from an unmanned aerial vehicle (UAV) during the AROMAT campaign, Atmospheric Measurement Techniques, 11, 551–567, https://doi.org/10.5194/amt-11-551-2018, 2018.

- Poraicu, C., Muller, J.-F., Stavrakou, T., Fonteyn, D., Tack, F., Deutsch, F., Laffineur, Q., Van Malderen, R., and Veldeman, N.: Cross-evaluating WRF-Chem v4.1.2, TROPOMI, APEX, and in situ NO2 measurements over Antwerp, Belgium, Geoscientific Model Development, 16, 479–508, https://doi.org/10.5194/gmd-16-479-2023, 2023.

- Seinfeld, J. H. and Pandis, S. N.: Atmospheric Chemistry and Physics: From Air Pollution to Climate Change, 3rd Edition, John Wiley and Sons, New York, ISBN 978-1-118-94740-1, 2016.

- Tack, F., Merlaud, A., Iordache, M.-D., Pinardi, G., Dimitropoulou, E., Eskes, H., Bomans, B., Veefkind, P., and Van Roozendael, M.: Assessment of the TROPOMI tropospheric NO2 product based on airborne APEX observations, Atmospheric Measurement Techniques, 14, 615–646, https://doi.org/10.5194/amt-14-615-2021, 2021.

---

## Author Comment (AC2)

We thank the referee for the thorough review of our manuscript, as well as for the careful evaluation and constructive suggestions. Referee comments are reproduced in black, and our replies are given in blue.

*This review is for egusphere-2025-3533, titled, Validation of TROPOMI and WRF-Chem NO2 across seasons using SWING+ and surface observations over Bucharest. The authors conduct a modeling study using WRF-Chem during specific days over Bucharest corresponding with research flights and validate them using airborne (SWING+) and ground-based data. WRF-Chem is then used as an intercomparison platform between SWING+ and TROPOMI to validate the satellite retrieval. Overall, the authors have done a thorough analysis, but this manuscript would only be suitable for publication after some more minor revisions addressing the comments and questions below but may range to a major revision depending on findings related to clouds, the SWING+ retrieval reference uncertainty, and doing a direct comparison between the SWING+ measurements and TROPOMI.*

*General comments:*

*Please clarify in the manuscript that this is analysis of tropospheric columns rather than total columns. This distinction may be needed in the validation section at the end of the paper as well.*

We added explicit mentions of tropospheric NO$_2$ columns in both sections.

*More detail is required on the SWING+ retrieval inputs and assumptions.*

- *What are the assumptions in albedo, a priori profiles, clouds, etc.? Is WRF-Chem used as a prior?*

We added the following information to Section 2.2.3.

"For radiometrically calibrated instruments such as APEX (Tack et al., 2019), surface reflectance can be retrieved through atmospheric correction of at-sensor radiance. However, for most airborne instruments (e.g., SWING+, AirMAP, Spectrolite; Tack et al., 2019), such calibration is not available. For SWING+, the albedo is therefore derived from MODIS surface properties, providing black-sky albedo at 470 nm (MCD43A3 v006; Schaaf and Wang, 2015) interpolated to each airborne pixel. The a priori profile is a well-mixed NO$_2$ box profile constrained by the ERA5 planetary boundary layer (PBL) height, under the assumption that the large majority of NO$_2$ resides within the PBL. Clouds are not considered, as all flights are conducted under cloud-free conditions, which is a strict requirement during flight planning."

We also added a footnote noting that more details on the SWING+ retrieval algorithm will be provided in a dedicated publication: *Airborne-Based Assessment of the TROPOMI Tropospheric NO$_2$ Product Across Multiple Campaigns* (F. Tack, A. Merlaud, T. Ruhtz, A. Nemuc, S. Iancu, D. Schuettemeyer, and M. Van Roozendael).

- *It is also contradicting to state that the reference is a daily average but then to say it's over a clean area. Both cannot be true.*

We agree that this wording was misleading. We have revised the entire paragraph to address this remark as well as some of the remarks mentioned below.

"AMFs are computed using the uvspec/DISORT radiative transfer model (Mayer and Kylling, 2005), with a relative uncertainty of 15.2% across the dataset. SCD$_{ref}$ represents a residual correction that accounts for the NO$_2$ amount present in the instrument reference spectrum. The reference spectrum is updated for each flight and calculated as the average of 30 spectra recorded over a clean area. The residual correction, associated with the average spectrum, was then estimated using interpolated SCD NO$_2$ data from TROPOMI (Veefkind et al., 2012). SCD$_{ref}$ values range from 0.5 to 2.1 × 10$^{15}$ molec. cm$^{-2}$, with an uncertainty estimated at 100%, yielding an error of 0.2-1.01 × 10$^{15}$ molec. cm$^{-2}$ after division by the AMF. Averaged per flight day, the DSCD uncertainty ranges from 1.4-2.5 × 10$^{15}$ molec. cm$^{-2}$, reducing to 0.5-1.6 × 10$^{15}$ molec. cm$^{-2}$ once divided by the AMF. The combined contributions of the AMF, SCD$_{ref}$, and DSCD yield a total VCD uncertainty of 0.9-1.9 × 10$^{15}$ molec. cm$^{-2}$. Lower uncertainties correspond to lower VCDs observed in spring and summer, while higher uncertainties are associated with elevated columns in fall and winter."

- *How is the reference amount estimated to add to the DSCD? It looks like some days may have reference issues in looking at the results.*

The estimation procedure, detailed in our response to the previous comment, is systematically repeated for each flight and carries a conservative uncertainty of 100% in the resulting value. This uncertainty is considered in the validation of TROPOMI NO$_2$ tropospheric products.

- *How are these airborne datasets from SWING+ validated?*

Several validation campaigns of SWING instruments have been conducted with larger airborne imagers and ground-based measurements, including a campaign over Bucharest (AROMAT 2015). These validation studies (Tack et al., 2019, Merlaud et al., 2020) are already referenced in the first paragraph of Sect 2.2.3. We added to this paragraph the information that these validation studies also included ground-based measurements.

"Initially designed for operations onboard an unmanned aerial vehicle (UAV) (Merlaud et al., 2018), SWING instruments have since been deployed on crewed aircraft for

validation flights alongside ground-based DOAS instruments and larger airborne imagers over Berlin (Tack et al., 2019) and Bucharest (Merlaud et al., 2020)."

- *How do you come to the uncertainty estimates around line 215?*

The uncertainty estimates are obtained by dividing the error on the slant column by the AMF, following standard uncertainty propagation. We have revised the text to clarify this procedure, as presented above.

- *Are these data cloud filtered? There do not appear to be gaps due to clouds in the maps. Validation over cloudy scenes would not be accurate.*

All flights were conducted under cloud-free conditions. Therefore, no data gaps are present. This has been clarified in the revised paragraph above.

*The paper uses WRF-Chem as a platform for intercomparison hitting upon understanding the challenge of vertical sensitivity and applying the averaging kernel to WRF-Chem and that the averaging kernel is different for aircraft and satellite. This is okay. However, this paper would greatly benefit with a direct comparison between SWING+ and TROPOMI rather than using the bias-corrected WRF-Chem columns as WRF-Chem columns will have a lot of uncertainty related to spatial gradients even with bias correction. This is the biggest source of uncertainty in this analysis even with the linear bias corrections. In addition to the analysis done already, this work would improve by adding the SWING+/TROPOMI comparison followed by discussion of the benefit of both techniques. The benefit of the direct comparison would be that the spatial information and magnitude of the column should originate from SWING+, not have an introduction of any spatial biases from WRF-Chem.*

- *There needs to be discussion on how the datasets are temporally matched because the linear corrections are also spanning time. This will be even more important with the direct comparisons of SWING+/TROPOMI.*

- *If the direct comparison is not done, the authors need to make a clearer case as to why they chose to validate TROPOMI with this technique. The model does have the benefit of filling the temporal gaps as noted in the introduction, but this benefit is not used in this analysis since its only during flight days.*

We thank the referee for highlighting aspects of our methodology that require clarification. We provide here a more detailed justification and have updated the manuscript where necessary. A direct SWING+/TROPOMI comparison is not included, as the presented methodology is designed to stand independently and focus on the model application. However, such a comparison will be explored in a dedicated forthcoming publication: *Airborne-Based Assessment of the TROPOMI Tropospheric $NO_2$ Product Across Multiple Campaigns* (F. Tack, A. Merlaud, T. Ruhtz, A. Nemuc, S.

Iancu, D. Schuettemeyer, and M. Van Roozendael), which will also present additional measurements from multiple campaigns.

Our choice of intercomparison method was motivated by two key advantages: it explicitly accounts for the vertical sensitivities of SWING+ and TROPOMI, and it minimizes the effect of temporal sampling differences. These features are fully exploited by applying the averaging kernels of both instruments and evaluating model columns within 5 minutes of each observation (Sections 2.2.3 and 2.2.4), ensuring temporal matching.

We agree that using the model as an intercomparison platform may introduce additional uncertainties. Conversely, direct comparisons avoid model-related errors but remain sensitive to differences in vertical sensitivity and to temporal gaps of up to one hour for SWING+. These gaps can introduce errors of up to $4 \times 10^{15}$ molec. $cm^{-2}$ in the $NO_2$ VCD (Merlaud et al., 2020). Consequently, both approaches may give rise to biases. We agree that comparing these approaches is of general interest and discuss them when reviewing results from Tack et al. (2021) and Poraicu et al. (2023) in Section 4. Nevertheless, our intercomparison approach is robust on its own, provided that model uncertainties are carefully assessed. In this study, this is achieved as follows:

- Model performance evaluation: Comparisons with in situ meteorological and chemical measurements (Sections 3.1) provide an initial consistency check, while comparisons with airborne observations (Section 3.2) allow us to exclude one date (22/11/2021) with unreliable spatial patterns.
- Bias evaluation of the model: Linear regressions are used to match daily SWING+ magnitudes and adjust spatial gradients at first order by estimating a concentration-dependent model bias (Section 3.3.1). The uncertainties in the regression results are also quantified.

We did not extend the TROPOMI validation to non-flight days because the model bias varies from day to day. Outside SWING+ flight days, we cannot estimate this bias nor its associated uncertainty with sufficient reliability.

To better reflect the need to assess model performance, we have modified a paragraph in the introduction as follows:

"Additionally, using a CTM such as WRF-Chem enables a quantitative comparison between SWING+ and TROPOMI products by bridging temporal lags and accounting for the vertical sensitivities of both instruments, using their averaging kernels. This method was applied by Zhu et al. (2016, 2020) for HCHO over the Southern United States and California, and by Poraicu et al. (2023) for $NO_2$ over the Antwerp region in Belgium. We revisit this intercomparison method in the present study by exploiting the large number of flight measurement days and explicitly propagating measurement errors. Unlike in a direct comparison, the intercomparison may also be affected by model errors.

Therefore, we use the assessment of the model against surface meteorological and chemical measurements, as well as airborne SWING+ observations, as a consistency check of model performance and to identify poorly performing simulation days before proceeding to TROPOMI validation."

More details on model error treatment (random vs systematic) are also provided in response to the specific comment referring to Line 435 below.

*The title implies that TROPOMI is validated with SWING+ and surface observations, but this is not the case as TROPOMI is validated with bias corrected WRF-Chem.*

We first use surface and airborne measurements to validate WRF-Chem and subsequently correct the WRF-Chem columns to account for SWING+ magnitudes and spatial gradients (to first order, as explained above). Therefore, we believe that the title appropriately reflects the content of our manuscript.

**Specific comments:**

*Line 40-41: The bias values noted do not appear in the report referenced. The latest says 13% and -40% for the bias values. Please check the references for these values.*

They appear at page 70 of the April 2018 – February 2025 edition ([https://s5p-mpc-vdaf.aeronomie.be/ProjectDir/reports//pdf/S5P-MPC-IASB-ROCVR-26.01.00_FINAL.pdf](https://s5p-mpc-vdaf.aeronomie.be/ProjectDir/reports//pdf/S5P-MPC-IASB-ROCVR-26.01.00_FINAL.pdf)).

*Line 95: Please clarify, is the model span up to 20km? or above 20km?*

We adapted the sentence as follows:

"The vertical grid of the model comprises 44 levels, reaching altitudes up to ca. 20 km."

*Line 96: It would be helpful to have the flight date table introduced here.*

Done.

*Line 98: The justification for the 3-hour spin-up time because Bucharest is UTC+3 is not a scientifically sound reason. Also, a 3-hour spin-up time seems incredibly short. Is there literature to support this?*

This choice is evaluated in the surface measurements analysis in Section 3.1, where the consistency of our results through the two-day periods indicates that our selected spin-up time is appropriate. Note that a longer spin-up is used for aircraft and satellite comparisons. The sentence has been revised for clarity:

"This setup allows for comparisons with in situ measurements over a two-day period (including the day preceding the flight and the flight day itself), with a spin-up time of 3 or 4 hours (18:00 UTC is 20:00 or 21:00 LT in Bucharest, depending on daylight saving

time). For comparisons with airborne and satellite measurements, the spin-up time exceeds 37 hours."

*Line 105: Define WPS*

WPS is defined at the beginning of Sect. 2.1.1.

*Line 124-125: The sentence starting with 'Its justification...' needs to be moved up to the second sentence of the paragraph.*

Done.

*Figure 3 and Table 2 do not add much helpful information to this analysis. Consider removing for a supplement or making clearer why it is needed within the analysis.*

The content of the figure has been converted into a table, which is now presented in Supplement 2.

*Line 163: Are these ground-based chemiluminescent measurements molybdenum converters or a different type? The correction factor in the literature is specific for molybdenum.*

The RNCMA network uses chemiluminescence instruments with molybdenum converters (Thermo Fisher model 42i) at each of the considered stations (https://www.calitateaer.ro/public/monitoring-page/). We added this information in the second paragraph of Sect. 2.2.2.

*Line 203: 'hovered' implies flying something like a helicopter or drone. Consider rewording to 'flew over' or 'operated'.*

We replaced "hovered" with "flew over".

*Line 260: Need more detail on 'insights'.*

The phrase "...insights gained from the first step" has been replaced with "...biases evaluated in the first step," and additional details regarding the linear regression have been included in the next paragraph.

"Both parametric and robust linear regression methods (Theil-Sen estimator; Theil (1950); Sen (1968)) are tested, with the latter suppressing the impact of outliers. Importantly, the resulting linear corrections not only adjust mean column magnitudes but also modify spatial gradients to first order, as the bias is estimated as a concentration-dependent quantity."

*Line 275-276: define MB and RMSE in text.*

Done.

*Section 3.1.2: The NO2 in this analysis is NO2\* or NO2? They appear to be used interchangeably but should be consistent throughout the text. It was reviewed assuming NO2\* throughout, so could the low bias in NO2\* be due to other assumptions in the model for the NOz species which may be not represented well in the model?*

We revised Section 3.1.2 to clarify that the discussion refers specifically to surface concentrations of $NO_2$*. The reviewer is correct that biases in other $NO_z$ species may contribute to the simulated bias in $NO_2$*. However, we expect the bias in $NO_2$* to qualitatively reflect the bias in $NO_2$ itself for the following reasons.

- PAN has a relatively long lifetime (from hours to tens of days; Seinfeld and Pandis, 2016) and is sensitive to boundary conditions. However, reported PAN mixing ratios in European urban environments are of order ≤ 1 ppb (Kahn et al., 2017), typically an order of magnitude lower than $NO_2$ (Seinfeld and Pandis, 2016). PAN is therefore not expected to dominate the $NO_2$* expression.

- $HNO_3$ is formed primarily through oxidation of $NO_2$. Excessive $NO_2$ oxidation in the model would lead to a negative bias in $NO_2$ and a positive bias in $HNO_3$. Owing to the factor of 0.35 applied to $HNO_3$ mixing ratio in the formula of Lamsal et al. (2008), this still results in a net negative bias in $NO_2$*, though smaller than that in $NO_2$ itself. Conversely, insufficient oxidation produces the opposite effect. $HNO_3$ has a shorter atmospheric lifetime than PAN due to rapid deposition and aerosol uptake (Seinfeld and Pandis, 2016), making it less sensitive to boundary conditions. Nonetheless, overestimating its sinks in the model can reduce the inferred $NO_2$*, and vice versa. This effect is, however, limited by the factor of 0.35 too.

- Alkyl nitrates are expected to be present at low mixing ratios (typically of the order of 1 ppb in cities) and represent only a minor fraction (~10%) of $NO_y$ in polluted continental environments (Perring et al., 2013).

*Section 3.2.1: It appears that the background NO2 for SWING+ is around zero rather than a realistic background value. Is this offset because SCDref is not added to the slant column or why is it so low? It may compensate for the offset in the peaks. It looks like the next flight has a more reasonable background value.*

As explained above, the residual slant column is systematically estimated for each flight day using a reference spectrum, resulting in flight-specific values. A conservative uncertainty of 100% is consistently applied and accounted for in the TROPOMI validation.

*Line 363-366: How are all flight objectively screened for these thermal instabilities? What metrics are used to screen the data outside it not agreeing with the model?*

We carefully examined all flight measurements and identified cases exhibiting suspiciously high background values comparable in magnitude to the city plume, as explained in Sect. 3.2.2. Two flights were retained for further investigation: 10/07/2021 and 30/06/2022. For these cases, we inspected the fitted *Resol* parameter from the QDOAS analysis of SWING+ products, which revealed thermal instabilities associated with altitude-induced temperature changes. The *Resol* parameter accounts for small differences in spectral resolution between the reference and analysed spectra and is equal to zero when the resolutions are identical. In practice, we removed data segments for which *Resol* fell below approximately −0.02 for a prolonged period, which coincidentally occurred from the start of the flight until 13:24 LT for both flights (see the figure below for 30/06/2022).

[Figure]

*Line 386:*

1. *Justify why a factor of 1.5 was chosen objectively using the noted statistics.*

2. *How are the statistics different if the factor of 1.5 is not applied? The writing implies this was also done but it is not shown.*

We revised the following paragraph in Section 3.2.3.

"The comparable numbers of days with either positive (7) or negative (10) biases in Table 6 suggest a balanced model behavior on average. The small overall bias across all selected dates (MB of $0.5 \times 10^{15}$ molec. cm$^{-2}$ and RB of 13%), along with the underestimation in surface $NO_2$* found in Sect. 2.2.2 (MB of -8 µg m$^{-3}$ and RB of -33%), provides a retrospective justification for increasing the CAMS-REG anthropogenic $NO_x$ emissions by a factor of 1.5, as proposed in Sect. 2.1.2.

The small overall model bias against SWING+ reflects compensating seasonal biases of opposite sign, indicating that a temporally varying scaling factor for $NO_x$ emissions may be more realistic. However, while finer, day-specific adjustments based on the column evaluations in Table 6 could be considered, they would likely introduce abrupt and

potentially unrealistic temporal variations in emissions, e.g., in November 2021, when the mean model bias ranges from -5% to +125% across different days. This variability may reflect the fact that, in addition to emission uncertainties, the model daily performance (e.g., chemistry and transport) on a limited set of days can strongly influence seasonal statistics, particularly in winter and fall, whereas spring and summer appear more consistent."

Moreover, we include a table in Supplement 3 comparing runs with and without the factor of 1.5 applied to CAMS-REG v7.0 $NO_x$ emissions for a selection of four dates (one per season).

*Line 399-400: Could the seasonal difference in bias be due individual flight days and issues with either the model or retrieval? Maybe boundary conditions in the model? Or perhaps the SCDref amount in the SWING+ retrieval? It appears the bias would be from 23/12/2021 in winter at least. Maybe 5/11/2021 for fall? This seems more realistic than a vertical mixing issue.*

We agree with the referee that individual days can have a strong influence on seasonal statistics in Table 6, particularly in winter and fall, whereas spring and summer appear more consistent. We added this information in the revised paragraph shown in response to the previous comment. Day-to-day variability in the reference slant column density ($SCD_{ref}$) also has a significant impact on daily biases. In contrast, boundary conditions are prescribed far from Bucharest (at least ~300 km from the city) and are therefore expected to play only a minor role in simulated $NO_2$ levels over the urban area. However, it seems to us that difficulties in accurately representing vertical mixing, a well-known challenge in WRF-Chem simulations (e.g., Poraicu et al., 2023; Kuhn et al., 2024), may provide a plausible explanation for the observed imbalance between surface and column $NO_2$.

*Line 424-425: While the true atmospheric vertical profile is not known, you can use the same assumption in both SWING+ and TROPOMI using WRF-Chem which would allow the two datasets to be intercompared. The text does not even share which a priori profile is used for SWING+.*

Information on the a priori profile used for SWING+ has been added in Sect. 2.2.3.

Using WRF-Chem profiles to adapt both SWING+ and TROPOMI retrievals is equivalent to applying their averaging kernels to WRF-Chem columns (Douros et al., 2023). However, even after this adaptation, a direct comparison of adapted TROPOMI with SWING+ would still be affected by their acquisition-time offset, introducing errors of up to $4 \times 10^{15}$ molec. cm$^{-2}$ (Merlaud et al., 2020), which the intercomparison approach is designed to minimize.

*Line 435: 'Because most the uncertainty is due to the TROPOMI columns...'; this statement does not seem valid as so many factors go into the bias corrected WRF-*

*Chem columns and the WRF-Chem columns themselves likely have a random uncertainty exceeding this value which should not decrease due to a linear bias correction. I think this number comes from the mean bias but it doesn't account for the random uncertainty in WRF-Chem when comparing to TROPOMI and SWING+ products.*

This line refers to the random uncertainty and has been corrected for clarity:

"Because most of the random uncertainty is due to the TROPOMI columns, ..."

We agree that WRF-Chem columns are generally affected by both systematic and random errors. However, over the short time span of a flight/satellite overpass and over an urban area the size of Bucharest, we expect model errors to correlate from pixel to pixel. In Section 3.3.1, we therefore treat model errors as primarily systematic and assume they are captured by a column-dependent linear relationship that varies from day to day. Two features of our method further limit the impact of any remaining spatial variability on the intercomparison:

- WRF-Chem columns are averaged to TROPOMI resolution, which smooths small-scale discrepancies.
- Linear regressions, including a robust Theil-Sen estimator (Theil (1950); Sen (1968)), are used instead of pointwise comparisons, which suppress the influence of outliers in the datasets.

The WRF-Chem biases derived in Section 3.3.1 are occasionally significant and reflect the magnitude of the model error, which we explicitly account for. These biases also inherit some random uncertainty from the precision of LR1, but this contribution is reduced by the daily sample size and by the regridding to TROPOMI resolution. As a result, the bias-corrected dataset has a propagated random component smaller than the pixel-wise precision error of TROPOMI.

In response to this discussion, we have included two additional pieces in Section 2.2.4 that describe in detail the treatment of model errors.

"Although model errors arise from both random and systematic sources, they are expected to be correlated from pixel to pixel within the short time window (typically less than 2 hours; see Table 1) and small spatial domain (Bucharest surroundings). These correlated errors are therefore treated as systematic and identified with the model bias, which is allowed to vary from one flight day to the next. Any remaining random component of the model error is further reduced through regridding to the TROPOMI resolution."

"Both parametric and robust linear regression methods are tested, with the latter suppressing the impact of outliers (Theil-Sen estimator; Theil (1950); Sen (1968)). Importantly, the resulting linear corrections not only adjust mean column magnitudes

but also modify spatial gradients to first order, as the bias is estimated as a concentration-dependent quantity."

*Line 448: Does the bias correction of the WRF-Chem columns account for the random error or does it just carry it over and can be represented by the random error in LR2? I think the first sentence may need to be reworded.*

The sentence has been adapted as follows:

"The bias-corrected WRF-Chem columns $\Omega^{bc}_{W,T}$ carry the random uncertainty of the SWING+ columns $\sigma_{S,\,rand}$ because LR1 propagates it through the regression."

*Line 469: explain why 05/01/2022 is more reliable.*

We added a note in Sect. 3.3.2:

"The flight day of 23/12/2021 shows less convincing results (Table 6) and is characterized by consistently high modeled background values (see Supplement 4), which may be due to inaccurate initial or boundary conditions for $NO_x$ species, oxidant concentrations, and/or heterogeneous chemistry on aerosols."

This last possibility is hinted at by a significant difference in aerosol optical depth (AOD) values measured at the INOE station in Măgurele on 23/12/2021 compared with 05/01/2022, as shown in the plots below (top: 23/12/2021, bottom: 05/01/2022).

[Figure]

[Figure]

**References**

- Douros, J., Eskes, H., van Geffen, J., Boersma, K. F., Compernolle, S., Pinardi, G., Blechschmidt, A.-M., Peuch, V.-H., Colette, A., and Veefkind, P.: Comparing Sentinel-5P TROPOMI NO2 column observations with the CAMS regional air quality ensemble, Geoscientific Model Development, 16, 509–534, https://doi.org/10.5194/gmd-16-509-2023, 2023.

- Khan M.A.H., Cooke M.C., Utembe S.R., Archibald A.T., Derwent R.G., Jenkin M.E., Leather K.E., Percival C.J., Shallcross D.E.: Global Budget and Distribution of Peroxyacetyl Nitrate (PAN) for Present and Preindustrial Scenarios, Int J Earth Environ Sci 2: 130, https://doi.org/10.15344/2456-351X/2017/130, 2017.

- Kuhn, L., Beirle, S., Kumar, V., Osipov, S., Pozzer, A., Bosch, T., Kumar, R., and Wagner, T.: On the influence of vertical mixing, boundary layer schemes, and temporal emission profiles on tropospheric NO2 in WRF-Chem – comparisons to in situ, satellite, and MAX-DOAS observations, Atmospheric Chemistry and Physics, 24, 185–217, https://doi.org/10.5194/acp-24-185-2024, 2024.

- Mayer, B. and Kylling, A.: Technical note: The libRadtran software package for radiative transfer calculations - description and examples of use, Atmospheric Chemistry and Physics, 5, 1855–1877, https://doi.org/10.5194/acp-5-1855-2005, 2005.

- Merlaud, A., Tack, F., Constantin, D., Georgescu, L., Maes, J., Fayt, C., Mingireanu, F., Schuettemeyer, D., Meier, A. C., Schonardt, A., Ruhtz, T., Bellegante, L., Nicolae, D., Den Hoed, M., Allaart, M., and Van Roozendael, M.: The Small Whiskbroom Imager for atmospheric compositioN monitoriNG (SWING) and its operations from an unmanned aerial vehicle (UAV) during the AROMAT campaign, Atmospheric Measurement Techniques, 11, 551–567, https://doi.org/10.5194/amt-11-551-2018, 2018.

- Merlaud, A., Belegante, L., Constantin, D.-E., Den Hoed, M., Meier, A. C., Allaart, M., Ardelean, M., Arseni, M., Bosch, T., Brenot, H., Calcan, A., Dekemper, E., Donner, S., Dorner, S., Balanica Dragomir, M. C., Georgescu, L., Nemuc, A., Nicolae, D., Pinardi, G., Richter, A., Rosu, A., Ruhtz, T., Schonhardt, A., Schuettemeyer, D., Shaiganfar, R., Stebel, K., Tack, F., Nicolae Vajaiac, S., Vasilescu, J., Vanhamel, J., Wagner, T., and Van Roozendael, M.: Satellite validation strategy assessments based on the AROMAT campaigns, Atmospheric Measurement Techniques, 13, 5513–5535, https://doi.org/10.5194/amt-13-5513-2020, 2020.

- Perring, A. E., Pusede, S. E., and Cohen, R. C.: An observational perspective on the atmospheric impacts of alkyl and multifunctional nitrates on ozone and secondary organic aerosol; Chem. Rev., 113, 5848-5870, https://doi.org/10.1021/cr300520x, 2013.

- Poraicu, C., Muller, J.-F., Stavrakou, T., Fonteyn, D., Tack, F., Deutsch, F., Laffineur, Q., Van Malderen, R., and Veldeman, N.: Cross-evaluating WRF-Chem v4.1.2, TROPOMI, APEX, and in situ NO2 measurements over Antwerp, Belgium, Geoscientific Model Development, 16, 479–508, https://doi.org/10.5194/gmd-16-479-2023, 2023.

- Schaaf, C. and Wang, Z.: MCD43A3 MODIS/Terra+Aqua BRDF/Albedo Daily L3 Global - 500m V006, NASA Land Processes Distributed Active Archive Center, https://doi.org/10.5067/MODIS/MCD43A3.006, 2015.

- Seinfeld, J. H. and Pandis, S. N.: Atmospheric Chemistry and Physics: From Air Pollution to Climate Change, 3rd Edition, John Wiley and Sons, New York, ISBN 978-1-118-94740-1, 2016.

- Sen, P. K.: Estimates of the Regression Coefficient Based on Kendall's Tau, Journal of the American Statistical Association, 63, 1379–1389, https://doi.org/10.1080/01621459.1968.10480934, 1968.

- Tack, F., Merlaud, A., Meier, A. C., Vlemmix, T., Ruhtz, T., Iordache, M.-D., Ge, X., van der Wal, L., Schuettemeyer, D., Ardelean, M., Calcan, A., Constantin, D., Schonhardt, A., Meuleman, K., Richter, A., and Van Roozendael, M.:

Intercomparison of four airborne imaging DOAS systems for tropospheric NO2 mapping – the AROMAPEX campaign, Atmospheric Measurement Techniques, 12, 211–236, https://doi.org/10.5194/amt-12-211-2019, 2019.

- Tack, F., Merlaud, A., Iordache, M.-D., Pinardi, G., Dimitropoulou, E., Eskes, H., Bomans, B., Veefkind, P., and Van Roozendael, M.: Assessment of the TROPOMI tropospheric NO2 product based on airborne APEX observations, Atmospheric Measurement Techniques, 14, 615–646, https://doi.org/10.5194/amt-14-615-2021, 2021.

- Theil, H.: A rank-invariant method of linear and polynomial regression analysis III, Nederl. Akad. Wetensch. Proc., 53, 1397–1412, https://www.nrc.gov/docs/ML1330/ML13304B799.pdf, 1950.

- Veefkind, J., Aben, I., McMullan, K., Forster, H., de Vries, J., Otter, G., Claas, J., Eskes, H., de Haan, J., Kleipool, Q., van Weele, M., Hasekamp, O., Hoogeveen, R., Landgraf, J., Snel, R., Tol, P., Ingmann, P., Voors, R., Kruizinga, B., Vink, R., Visser, H., and Levelt, P.: TROPOMI on the ESA Sentinel-5 Precursor: A GMES mission for global observations of the atmospheric composition for climate, air quality and ozone layer applications, Remote Sensing of Environment, 120, 70–83, https://doi.org/https://doi.org/10.1016/j.rse.2011.09.027, the Sentinel Missions - New Opportunities for Science, 2012.

- Zhu, L., Jacob, D. J., Kim, P. S., Fisher, J. A., Yu, K., Travis, K. R., Mickley, L. J., Yantosca, R. M., Sulprizio, M. P., De Smedt, I., Gonzalez Abad, G., Chance, K., Li, C., Ferrare, R., Fried, A., Hair, J.W., Hanisco, T. F., Richter, D., Jo Scarino, A.,Walega, J., Weibring, P., and Wolfe, G. M.: Observing atmospheric formaldehyde (HCHO) from space: validation and intercomparison of six retrievals from four satellites (OMI, GOME2A, GOME2B, OMPS) with SEAC4RS aircraft observations over the southeast US, Atmospheric Chemistry and Physics, 16, 13 477–13 490, https://doi.org/10.5194/acp-16-13477-2016, 2016.

- Zhu, L., Gonzalez Abad, G., Nowlan, C. R., Chan 1005 Miller, C., Chance, K., Apel, E. C., DiGangi, J. P., Fried, A., Hanisco, T. F., Hornbrook, R. S., Hu, L., Kaiser, J., Keutsch, F. N., Permar, W., St. Clair, J. M., and Wolfe, G. M.: Validation of satellite formaldehyde (HCHO) retrievals using observations from 12 aircraft campaigns, Atmospheric Chemistry and Physics, 20, 12 329–12 345, https://doi.org/10.5194/acp-20-12329-2020, 2020.